# S$^2$-DMs: SKIP-STEP DIFFUSION MODELS

## ABSTRACT

Diffusion models have emerged as powerful generative tools, rivaling GANs in sample quality and mirroring the likelihood scores of autoregressive models. A subset of these models, exemplified by DDIMs, exhibit an inherent asymmetry: they are trained over $T$ steps but only sample from a subset of $T$ during generation. This selective sampling approach, though optimized for speed, inadvertently misses out on vital information from the unsampled steps, leading to potential compromises in sample quality. We refer to this phenomenon as "asymmetric diffusion models". To address this issue, we present the S$^2$-DMs, which use an innovative $L_{skip}$, meticulously designed to reintegrate the information omitted during the selective sampling phase. The benefits of this approach are manifold: it notably enhances sample quality, is exceptionally simple to implement, necessitates minimal code modifications, and is flexible enough to be compatible with various sampling algorithms. The S$^2$-DMs achieves strong results on the CIFAR10 (32x32) and CelebA (64x64) datasets(e.g., FID scores of 8.01/6.41 in just 10 steps, surpassing the performance of DDIMs and PNDMs). Access to the code and additional resources is provided in material.

## 1 INTRODUCTION

Generative models, especially deep generative models, play a foundational role in the machine learning domain(Karras et al. (2020); Oord et al. (2016)). Architectures like Variational Autoencoders (VAEs; Kingma & Welling (2013)) and Autoregressive models(Van den Oord et al. (2016);Brown et al. (2020); Salimans et al. (2017)), Generative Adversarial Networks (GANs; Goodfellow et al. (2014); Yu et al. (2017); Hjelm et al. (2017);Fedus et al. (2018)), and Restricted Boltzmann Machines (RBMs; Hinton (2012)) have been at the forefront. VAEs, while providing a structured probabilistic framework, occasionally yield blurry samples. GANs, acclaimed for their prowess in generating high-resolution images, can face training instabilities(Adler & Lunz (2018); Gulrajani et al. (2017); Karras et al. (2019)). RBMs, though seminal, find themselves overshadowed by more recent architectures in scalability and performance. Against this backdrop, diffusion models, Denoising diffusion probabilistic models(DDPMs; Ho et al. (2020); Sohl-Dickstein et al. (2015);Song et al. (2020b)), have emerged as a compelling alternative, exhibiting unmatched capabilities in generating superior samples in diverse domains, from image synthesis to molecule design(Bengio et al. (2014)).

However, diffusion models do come with challenges. Their inherently slow sampling speed, driven by the multitude of necessary sampling steps, remains a significant concern. Recent research has honed in on this computational bottleneck, with the goal of optimizing the sampling process (Jolicoeur-Martineau et al. (2021), Nichol & Dhariwal (2021)). A significant breakthrough in this area is the Denoising Diffusion Implicit Models (DDIMs; Song et al. (2020a)). DDIMs utilize a subset sampling strategy, achieving faster performance by sampling from a smaller subset instead of the entire set of steps. This method, due to its omission of certain steps, is coined "skip-step sampling." Yet, this acceleration introduces an inconsistency between training and sampling. During training, the model undergoes every step, but during sampling, some steps are selectively skipped, posing a risk of information loss. Although DDIMs, with certain mathematical adjustments, have lessened the adverse effects of this approach compared to DDPMs, they haven't specifically addressed and optimized for the missing intermediate information. Consequently, this challenge persists, leading to the suboptimal performance of diffusion models during expedited sampling and hindering the generation of high-quality samples.

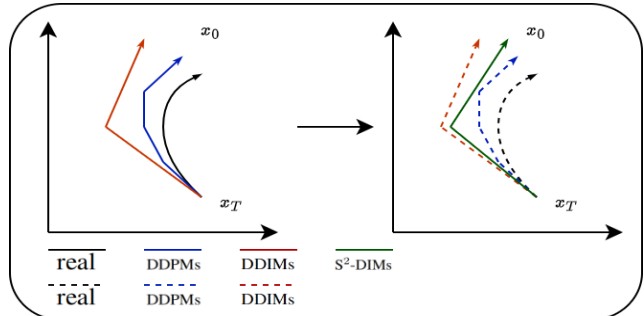

Figure 1: Sampling trajectory examples of DDPMs, DDIMs, and the $S^2$-DMs. The 'real' refers to the actual transition trajectories between two distributions. The left side of the figure depicts the sampling trajectories of different algorithms, where the dashed lines represent the trajectories of other algorithms, and the solid lines denote those of our proposed algorithm. The $x_T$ denotes the normal distribution, and $x_0$ represents the data distribution. The process involves sampling from the normal distribution and restoring it to the $x_0$ data distribution via various algorithmic trajectories.

Driven by these observations, our research proposes a method that integrates the selective sampling feature directly into the training process. In this manner, the model is primed to account for and adapt to the information that might be missed during sampling. This approach not only ensures remarkably quick sampling times but also preserves the quality and fidelity of the generated samples, striking a balance between efficiency and performance. During training, the original loss function is preserved. In parallel, a skip-step loss is introduced, measuring the discrepancy between the current step's prediction and the skip-step result. This skip-step loss is combined with the original loss using a weighted mechanism. As a result, insights from the skip-step are seamlessly incorporated during training, eliminating the need for extra adjustments during sampling (see Figure 1). This ensures the production of top-tier samples under consistent conditions.

Empirical results demonstrate that when the skip-step loss is incorporated into the loss objective, the $S^2$-DMs deliver state-of-the-art performance in unconditional generation on CIFAR10 (Krizhevsky et al. (2009)), outperforming DDIMs and PNDMs (Liu et al. (2022)) under comparable conditions. Importantly, our findings indicate that this method permits the model to train with fewer steps while still producing exceptional outcomes. The resulting FID score on CIFAR10 is an impressive 8.01 in just 10 steps. Similarly, its performance on CelebA surpasses that of DDIMs, registering an FID score of 6.41 in 10 steps. The $S^2$-DMs represent a leading-edge optimization technique addressing the disparity between training and sampling in diffusion models. Furthermore, it's noteworthy that while this method is particularly effective for DDIMs, we also evaluated the sampling algorithm of PNDMs and continued to see stellar results, underscoring the general applicability of our method to various diffusion models. The key **contributions** of this work are:

1. Innovative Skip-Step Loss: We introduce a trailblazing skip-step loss, embedding the selective sampling modality directly within the training process. This method empowers models to proactively navigate potential sampling information deficits, enhancing the quality of the samples.

2. Simplicity of Implementation: The $S^2$-DMs approach stands out not just for its efficacy but also its simplicity. With minimal code alterations required, it offers a convenient solution for both researchers and practitioners. Crucially, it's adaptable to a range of sampling algorithms.

3. Addressing Training-Sampling Disparity: Our research presents the inaugural method specifically designed to mitigate the inherent training and sampling mismatch in diffusion models. This strategy consistently showcases superior performance.

## 2 BACKGROUND

This study is based on DDPMs (Ho et al. (2020)) and DDIMs (Song et al. (2020a)), so a brief review is in order. DDPMs specifies a prior Markov forward diffusion process, which gradually adds noise

to the data over $T$ steps. Refer to the background description of Watson et al. (2021). Following the notation of (Ho et al. (2020)),

$$q(x_0, ..., x_T) = q(x_0) \prod_{t=1}^{n} q(x_t|x_{t-1}), \tag{1}$$

$$q(x_t|x_{t-1}) = \mathcal{N}(x_t|\sqrt{\alpha_t}x_{t-1}, 1 - \alpha_t I), q(x_t|x_0) = \mathcal{N}(x_t|\sqrt{\bar{\alpha}_t}x_{t-1}, 1 - \bar{\alpha}_t I), \tag{2}$$

where $q(x_0)$ represents the data distribution and $1 - \alpha_t$ signifies the variance of the Gaussian noise added at step $t$. For each $t$, we have $\alpha_t = 1 - \beta_t$ and $\bar{\alpha}_t = \prod_{s=1}^{t} \alpha_s$. To facilitate the transformation of noise back into data, DDPMs are trained to invert equation 1 with a model $p_\theta(x_{t-1}|x_t)$. This model is trained by optimizing a (possibly reweighted) evidence lower bound (ELBO).

$$E_q[D_{KL}[q(x_T|x_0)||p(x_T)] + \sum_{t=2}^{T} D_{KL}[q(x_{t-1}|x_t, x_0)||p_\theta(x_{t-1|x_t}] - \log p_\theta(x_0|x_1)]. \tag{3}$$

DDPMs explicitly select the model for parameterization as

$$p_\theta(x_{t-1}|x_t) = q(x_{t-1}|x_t, \frac{1}{\sqrt{\bar{a}_t}}(x_t - \sqrt{1 - \bar{a}_t}\epsilon_\theta(x_t, t)))$$
$$= \mathcal{N}(x_{t-1}|\frac{1}{\sqrt{\alpha_t}}(x_t - \frac{1 - \alpha_t}{\sqrt{1 - \bar{a}_t}}\epsilon_\theta(x_t, t)), \frac{1 - \bar{a}_{t-1}}{1 - \bar{a}_t}\beta_t I). \tag{4}$$

In this framework, optimizing the ELBO corresponds to minimizing denoising score matching goals as explained by Vincent (2011). Song et al. (2020a) introduced the DDIMs concept, a set of EL-BOs complemented by forward diffusion processes and sampling mechanisms. These ELBOs, having similar marginals as DDPMs, offer flexibility in determining posterior variances (Chen et al. (2020)). Song et al. (2020a) emphasized crafting alternative ELBOs using a subset of original timesteps $S \subset \{1, ..., T\}$ with consistent marginals. This results in $q_S(x_t|x_0) = q(x_t|x_0)$ for every $t$ in $S$, permitting faster sampling processes compatible with pre-trained models by integrating new timesteps. Their work also suggests the feasibility of creating a vast range of non-Markovian processes, denoted as $\{q_\sigma : \sigma \in [0, 1]^{T-1}\}$, with each $q_\sigma$ maintaining marginals aligned with the original progression.

$$q_\sigma(x_0, ..., x_t) = q(x_0)q(x_T|x_0) \prod_{t=1}^{T-1} q_\sigma(x_t|x_{t+1}, x_0), \tag{5}$$

and where the posteriors are defined as

$$q_\sigma(x_{t-1}|x_t, x_0) = \mathcal{N}(x_{t-1}|\sqrt{\bar{\alpha}_{t-1}} \left( \frac{x_t - \sqrt{1 - \bar{\alpha}_t}\epsilon_\theta}{\sqrt{\bar{\alpha}_t}} \right) + \sqrt{1 - \bar{\alpha}_{t-1} - \sigma^2}\epsilon_\theta, \sigma_t^2 I). \tag{6}$$

In their research, Song et al. (2020a) observed that the special case of employing all-zero variances, termed as DDIMs($\eta = 0$), persistently enhances the quality of samples in the short-step domain. When amalgamated with an apt choice of timesteps for assessing the modeled score function, known as strides, DDIMs($\eta = 0$) sets a new benchmark in the realm of few-step diffusion model sampling, especially with minimal inference step allocations. A pivotal advancement we bring is the enhancement of sample quality by introducing skip information (i.e., the aforementioned subset) during the training phase, ultimately establishing a novel diffusion model training paradigm. For a more comprehensive discussion on the $S^2$-DMs family, please refer to Section 3.

## 3 SKIP-STEP DIFFUSION MODELS

Acceleration approaches under DDIMs (Song et al. (2020a)) often employ skip-step sampling as a strategy for acceleration. However, this approach inherently introduces non-smooth denoising, leading to a potential decline in performance. This observation prompted us to re-evaluate the entire training and sampling workflow. Intriguingly, we identified an asymmetry between the training and sampling phases: the former proceeds in single steps, while the latter uses skip-step.

To enhance the efficacy of skip-step sampling, we devised a novel yet straightforward objective function for the training phase. By incorporating this new objective into our original loss function (Section 3.2), we achieved a symmetrical training effect. Ultimately, our model's design remains simple, and its performance meets our expectations (Section 4).

## 3.1 Asymmetry in Accelerated Sampling

Due to the slow sampling speed of diffusion models, extensive research has been conducted on acceleration algorithms for these models, with DDIMs being the most prominent. In the original paper, it was stated that a subset of the full $T$ steps of the diffusion model was selected. This subset forms an increasing sequence and is considerably shorter than the original $T$ steps, leading to a significant acceleration in the sampling process. Specifically, in its implementation, not all $T$ steps are sampled. Instead, 50-step and 100-step samplings are more prevalent, which are 10-20 times faster than the original 1000-step sampling. For instance, in the 100-step sampling, the model samples at every 10th step, maintaining equal intervals between each sample. This method is termed "skip-step sampling."

Clearly, there's an asymmetry between the behavior during training and sampling. During training, the model is trained across all diffusion steps. In contrast, during sampling, it samples only a subset of these steps using skip-step sampling. Consequently, information from intermediate steps is overlooked, inevitably leading to a decline in model performance. We term this the "asymmetric diffusion model."

In the subsequent sections, we will present a technique to integrate skip-step information during the training phase. This approach ensures that the trained model is more attuned to the skip-step sampling process, culminating in what we call the "Skip-Step Diffusion Models."

## 3.2 Training with Skip-Step Loss

We aim to introduce a novel skip-step loss function built upon the original one. The standard optimization function for diffusion models was presented in the DDPMs and subsequent diffusion models predominantly utilize this foundational loss function,

$$L_0 = \mathbb{E}_{t,x_0,\epsilon}||\epsilon - \epsilon_\theta(\sqrt{\alpha_t}x_0 + \sqrt{1-\alpha_t}\epsilon,t)||^2. \tag{7}$$

Initially, we assume that sampling is conducted every 10 steps, which aligns with the commonly used DDIMs setting. This configuration allows us to reduce the sampling from the original 1000 steps to just 100 steps. Here, the skip-step setting corresponds to the sampling time, denoted as skip=10(Subsequent experiments will explore the model performance with various skip values). We will now introduce skip-step information, and for this purpose, we define $\alpha_{skip}^t = \alpha_t \cdot \alpha_{t-1} \ldots \alpha_{t-9}$.

Then the role of $L_0$ is to enable the model to learn the information at each step, which corresponds to $q(x_t|x_{t-1})$. However, during the training phase, the corresponding sampling step is $p(x_{t-skip}|x_t)$. Hence, we consider it as a new skip-step loss function. During the training process of the model, we aim to make $q_\theta(x_t|x_{t-skip})$ as close as possible to $q(x_t|x_{t-1})$. By doing so, when sampling with skips, the model can produce outputs that are closely aligned with the corresponding positions, thereby enhancing the quality of the output. Their formulations are as follows:

$$q_\theta(x_t|x_{t-skip}) = \sqrt{\alpha_{skip}^t}x_{t-skip} + \sqrt{1-\alpha_{skip}^t}\epsilon_\theta, \tag{8}$$

$$q(x_t|x_{t-1}) = \sqrt{\alpha_t}x_{t-1} + \sqrt{1-\alpha_t}\epsilon. \tag{9}$$

We can adopt the approach of the original loss function. Since $\alpha_{skip}x_{t-skip}$ and $\sqrt{\alpha_t}x_{t-1}$ are numerically very close and can be ignored, and our goal is to have the model fit $\epsilon$, we can get the following equation:

$$L_{skip} = \mathbb{E}_\epsilon \left\| \sqrt{1-\alpha_t}\epsilon - \sqrt{1-\alpha_{skip}^t}\epsilon_\theta \right\|^2 \tag{10}$$

$$= \frac{1}{\sqrt{1-\alpha_t}}\mathbb{E}_{t,x_0,\epsilon} \left\| \epsilon - \frac{\sqrt{1-\alpha_{skip}^t}}{\sqrt{1-\alpha_t}}\epsilon_\theta \left(\sqrt{\alpha_t}x_0 + \sqrt{1-\alpha_t}\epsilon, t\right) \right\|^2. \tag{11}$$

Finally, to maintain consistency with the original loss function, we discarded the coefficients. Due to the significant variations in $\sqrt{1-\alpha_t}$, the scale of $L_{skip}$ can change dramatically, which could

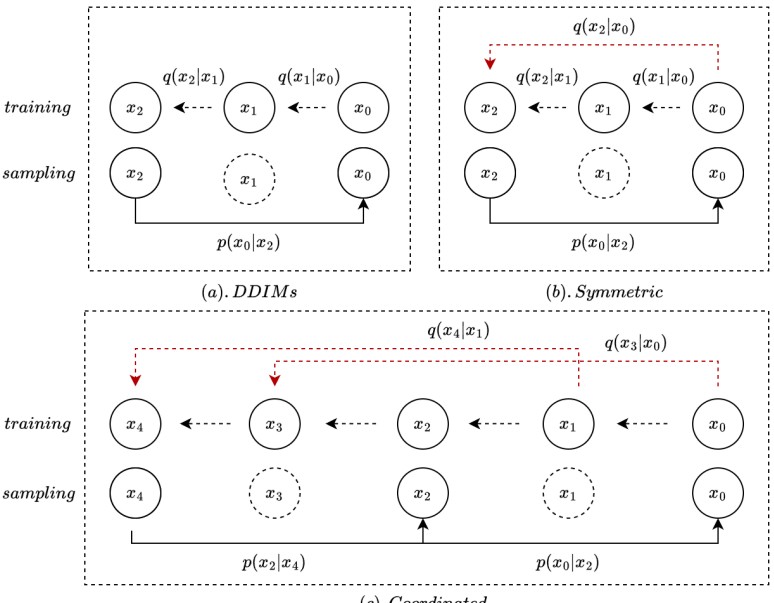

Figure 2: (a) Represents the asymmetric nature of DDIMs. Specifically, during the training phase, every step is trained, while the sampling phase skips certain steps. (b) Represents the symmetry of the $S^2$-DMs. During the training process, not only is every step trained, but to maintain consistency with the sampling phase, skip-step training is also conducted. (c) Represents the coordination of the $S^2$-DMs. That is, after incorporating skip-step training, the model becomes better adapted to skip-step sampling, yielding superior results. It exhibits its best performance within a specific symmetric interval, without the need for strict symmetric sampling. Once the training concludes, the sampling phase directly employs the step-skip approach, inherently encompassing the information of the intermediate skipped steps.

lead to model training instability. To ensure a more stable value, we replaced the denominator with the more stable $\sqrt{\alpha^t_{skip}}$.

$$L_{skip} = \mathbb{E}_{t,x_0,\epsilon} \left\| \epsilon - \frac{\sqrt{1-\alpha^t_{skip}}}{\sqrt{\alpha^t_{skip}}} \epsilon_\theta \left( \sqrt{\alpha_t}x_0 + \sqrt{1-\alpha_t}\epsilon, t \right) \right\|^2 . \tag{12}$$

Due to the modification of some weight values, it is essential to introduce an appropriate weight in the subsequent sections to ensure the system functions properly. Currently, the value of this loss function is stable and does not lead to collapse (see the experimental in section 4).

### 3.3 LOSS SCALING

Traditionally in the training of diffusion models, there is typically only one training objective, $L_0$. However, our proposed $S^2$-DMs introduce a new training objective, $L_{skip}$. This necessitates a reasonable weight to balance the two objectives. In subsequent experiments, we separately examined the values of the two losses and found that the average value of $L_{skip}$ is approximately 80-100 times larger than that of $L_0$. Therefore, when setting the weights, this scale difference must be taken into account. We assigned a weight of $\tau$ to $L_0$ and $(1-\tau)$ to $L_{skip}$. With $\tau$=0.99, we effectively balanced the scales of $L_0$ and $L_{skip}$, integrating them into a comprehensive training objective.

$$L = \tau L_0 + (1 - \tau)L_{skip}. \tag{13}$$

Finally, from the perspective of the training objective, its form is akin to a regularization term. The design motivation is indeed to provide information compensation for the skip-step diffusion model, thus constraining the model's trajectory. This is somewhat analogous to the idea of regularization.

### 3.4 TRAINING AND SAMPLING

Figure 2 highlights our method's core, illustrating states: symmetric and coordinated. By integrating skip-step data during training, the model gains a broader perspective, enhancing sampling performance. While the model may lean towards symmetry, it's not a prerequisite for optimal performance. Peak efficacy is seen when nearing a symmetric form, termed "coordinated". This aligns with the model utilizing both current and post-skip data for improved predictions. The model remains flexible, not restricted by the $skip$ parameter, allowing diverse step sampling.

Incorporating $L_{skip}$ during training doesn't introduce a new sampling method but modifies the diffusion model's traditional training approach. In essence, models following the original diffusion training can benefit from our method. Our tests, using different sampling algorithms on identically trained models, consistently matched our predictions (see Experiment 4).

In Algorithm 1 and 2, we illustrate the training and sampling procedures of the S$^2$-DMs. The training process, compared to the standard diffusion models, only involves an additional computation of $L_{skip}$. This computation is straightforward. In our code repository, one can see that only a few lines of the entire code were modified to achieve all changes, making it easy to implement and facilitating follow-up by other researchers. The sampling procedure follows the standard DDIMs sampling. As skip-step information was incorporated during the training, no modifications are required in the sampling process. This allows for the generation of higher-quality samples, making it user-friendly.

---

**Algorithm 1** S$^2$-DMs Training process.

1: **repeat**
2: $\quad x_0 \sim q(x_0)$;
3: $\quad t \sim Uniform(1, ..., T)$;
4: $\quad \epsilon \sim \mathcal{N}(0, I)$;
5: $\quad L_0 = \nabla_\theta ||\epsilon - \epsilon_\theta||^2$;
6: $\quad L_{skip} = \nabla_\theta ||\epsilon - \frac{\sqrt{(1-\alpha_{skip})}}{\sqrt{\alpha_{skip}}}\epsilon_\theta||^2$;
7: $\quad$ Take gradient descent step on:
$\quad\quad (1 - \tau) \cdot L_0 + \tau \cdot L_{skip}$;
8: **until** convergence is achieved

---

**Algorithm 2** S$^2$-DMs sampling with DDIMs.

1: **repeat**
2: $\quad x_T \sim \mathcal{N}(0, 1)$;
3: $\quad$ **for** $t = T, ..., 1$ **do**
4: $\quad\quad if\ t > 0 : \sigma \sim \mathcal{N}(0, I)$
$\quad\quad else : \sigma = 0$;
5: $\quad\quad x_{t-1} = \sqrt{\bar{\alpha}_{t-1}}\left(\frac{x_t - \sqrt{1-\bar{\alpha}_t}\epsilon_\theta}{\sqrt{\bar{\alpha}_t}}\right)$
$\quad\quad\quad + \sqrt{1 - \bar{\alpha}_{t-1} - \sigma^2}\epsilon_\theta + \sigma^2\epsilon$;
6: $\quad$ **end for**
7: **until** convergence is achieved

---

## 4 EXPERIMENTS

In this section, we demonstrate that the S$^2$-DMs outperform DDIMs(Song et al. (2020a)) and PNDMs(Liu et al. (2022)) in image generation with the same number of steps. The S$^2$-DMs requires fewer iterations to produce images of high quality. Moreover, the latent variables in the images generated by the S$^2$-DMs retain a high level of image features, allowing for interpolation within the latent space.

In consideration of computational resources, our experiments utilized the CIFAR10 dataset with a resolution of 32×32 and the CelebA dataset with a resolution of 64×64. For the training setup, we adopted the same architecture(He et al. (2016); Ronneberger et al. (2015); Kingma & Ba (2014)) as provided in the official DDIMs repository. We also ensured that all parameters were kept consistent, guaranteeing the reproducibility of our experiments. On the hardware front, both datasets were trained on two NVIDIA A100 GPUs. The CIFAR10 and CelebA datasets were trained for one day and two days respectively. Model performance was evaluated based on the FID(Heusel et al. (2017); Jolicoeur-Martineau et al. (2020)). Specifically, our evaluation method was also in strict accordance with the DDIMs repository, where we sampled 50,000 images and computed the FID against real images. To further ensure reproducibility, we fixed the random seed in our experiments, making all results replicable.

### 4.1 SAMPLE QUALITY AND ABLATION EXPERIMENT

In Table 1 and 2, we evaluate the quality of samples generated by models trained on the CIFAR10 and CelebA datasets, measured using the FID as the evaluation metric. We default to $skip = 10$ and

Table 1: FID scores for the S$^2$-DMs against baseline methods trained on CIFAR10(32x32) with the $L_{skip}$. DDIMs[1] and PNDMs was training for 600K/400K iterations. DDIMs[2] was trained for 800K/600K iterations. For the S$^2$-DMs(DDIMs) and S$^2$-DMs(PNDMs), we set $skip = 10$, while other parameters remained consistent, and it was trained for 600K/400K iterations.

| Models \ # samplesteps S | 10 | 20 | 50 | 100 | 200 | 500 | 1000 |
|---|---|---|---|---|---|---|---|
| DDIMs[1] | 18.32 | 11.67 | 8.07 | 6.33 | 5.38 | 4.59 | 4.54 |
| DDIMs[2] | 18.18 | 11.59 | 7.94 | 6.24 | 5.25 | 4.58 | 4.46 |
| S$^2$-DMs(DDIMs) | **15.63** | **9.88** | **6.75** | **5.61** | **4.87** | **4.30** | **4.21** |
| Other | | | | | | | |
| PNDMs | 13.67 | 7.61 | 4.87 | 3.99 | 3.67 | 3.56 | 3.42 |
| S$^2$-DMs(PNDMs) | **12.01** | **6.54** | **4.36** | **3.77** | **3.55** | **3.43** | **3.26** |

Table 2: FID scores for the S$^2$-DMs against baseline methods trained on CelebA(64x64) with the $L_{skip}$.

| Models \ # samplesteps S | 10 | 20 | 50 | 100 | 200 | 500 | 1000 |
|---|---|---|---|---|---|---|---|
| DDIMs[1] | 13.15 | 9.29 | 6.40 | 5.24 | 4.58 | 4.18 | 4.07 |
| DDIMs[2] | 13.12 | 9.25 | 6.34 | 4.63 | 4.18 | 4.15 | 4.06 |
| S$^2$-DMs(DDIMs) | **11.97** | **8.12** | **5.29** | **4.18** | **3.65** | **3.25** | **3.13** |
| Other | | | | | | | |
| PNDMs | 12.59 | 8.72 | 6.00 | 4.89 | 4.30 | 4.12 | 3.43 |
| S$^2$-DMs(PNDMs) | **11.40** | **7.58** | **4.94** | **3.91** | **3.38** | **3.01** | **2.94** |

compare our results with DDIMs. As expected, by incorporating skip-step information, the model is able to capture a broader scope of knowledge, leading to an improved sample quality. We observed that the S$^2$-DMs consistently produces higher quality samples than DDIMs and PNDMs across different sampling steps. Moreover, the advantage of the S$^2$-DMs becomes even more pronounced with shorter trajectories. This demonstrates that the diffusion model enhanced by our training algorithm contributes to the improvement of sample quality. Moreover, it is not limited to DDIMs sampling methods but is also applicable to other accelerated sampling approaches. Thus, other models only need a few lines of training code modifications to benefit from the performance boost this method offers, without any additional changes. An intriguing phenomenon we noticed, and is also depicted in Table 1, is that by adding $L_{skip}$, the model requires fewer training steps to achieve excellent performance. We speculate that the inclusion of skip-step information accelerates the model's convergence rate.

In Figure 3, we demonstrate the influence of different skip-step information on the model, conducted on the same datasets(more data details in Table 5). We adopt {50, 10, 2} as skip-step intervals for the model. As anticipated, incorporating more distant skip-step information allows the model to achieve a broader perspective in fewer generative trajectories, resulting in higher quality samples. However, we discovered that the relationship between skip-step information and the quality of the sampling step is not strictly symmetric. For instance, with $skip = 50$, one would expect the best performance at $step = 50$. Yet, the experiments do not consistently confirm this expectation: it holds true for CelebA but not for CIFAR10. This highlights the harmony and symmetry we mentioned. After adding skip-step information, the model's training and sampling processes become more harmonized, leading to superior sample quality. Still, there's a notion of symmetry where, within a symmetric interval, the quality of generated samples peaks.

In Figure 4 and 5, we showcase samples from the CIFAR10 and CelebA datasets generated by models with the same number of sampling steps but different architectures. For DDIMs and PNDMs, when the number of sampling steps is limited, the quality of the generated samples is inferior to those from the S$^2$-DMs. Furthermore, as the skip setting increases, the quality of the samples produced improves, with richer details in the generated images. For example, in CIFAR10, the second and fourth columns depict images of boats and cars respectively, indicating that the S$^2$-DMs can generate highly accurate images in just 10 steps, while DDIMs produces blurry images, requiring more steps to achieve high-quality results. Similarly, in CelebA, the third column showcases images

of a male subject. The $S^2$-DMs produce a clear hat for him, whereas DDIMs still renders a rather blurry hat. Moreover, the details generated by PNDMs are also less compared to those by the $S^2$-DMs. The difference is indeed substantial. This underscores that the $S^2$-DMs significantly improves sample quality with fewer sampling steps. This underscores the $S^2$-DMs's significant enhancement in sample quality with fewer sampling steps.

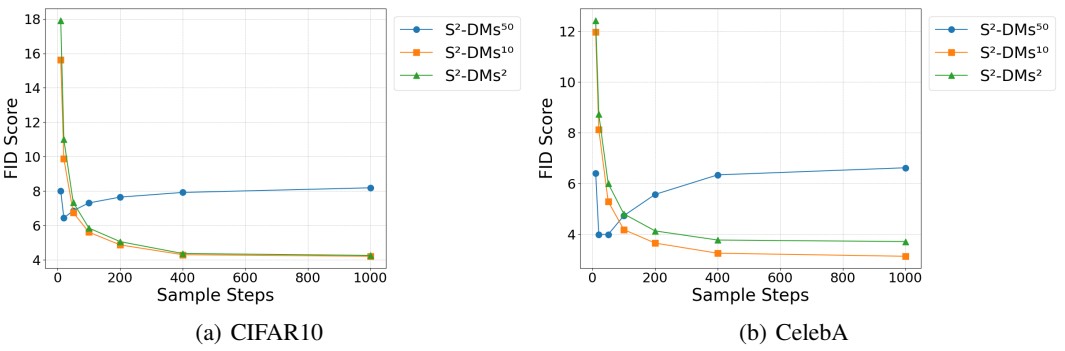

(a) CIFAR10          (b) CelebA

Figure 3: FID scores for the step ablation on CIFAR10 and CelebA. The impact of skip steps on the model was examined by varying the skip values among {50, 10, 2} based on DDIMs.

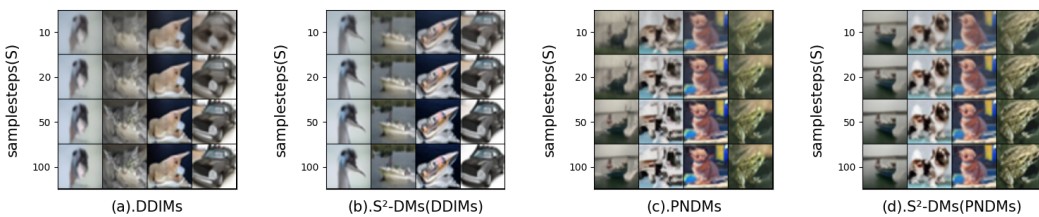

(a).DDIMs     (b).S²-DMs(DDIMs)     (c).PNDMs     (d).S²-DMs(PNDMs)

Figure 4: CIFAR10 and CelebA samples with difference models in {10, 20, 50, 100} steps.



(a).DDIMs     (b).S²-DMs(DDIMs)     (c).PNDMs     (d).S²-DMs(PNDMs)

Figure 5: CIFAR10 and CelebA samples with difference models in {10, 20, 50, 100} steps.

## 4.2 INTERPOLATION AND GENERATION CONSISTENCY

Given that the $S^2$-DMs is based on the deterministic generation process of DDIMs(Song et al. (2020a)) and PNDMs(Liu et al. (2022)), it also exhibits the semantic interpolation effects observed in implicit models(Mohamed & Lakshminarayanan (2016)), such as GANs. In Figure 6, we display the interpolation results of the $S^2$-DMs under different skip-step settings. From the figure, it can be discerned that simple interpolation in $x_T$ can lead to semantically meaningful interpolations between two samples. Moreover, the generated samples at $skip = 50$ demonstrate superior quality and finer details, as exemplified by the 5th to 7th images, where the light and shadow effects on the faces are also effectively reproduced. In contrast, at $skip = 2$, the model closely resembles the original DDIMs, resulting in a loss of sample detail. Additionally, the figure illustrates that even models trained with different skip-step settings, when conditioned on the same $x_T$ encoding, still produce fairly consistent samples.

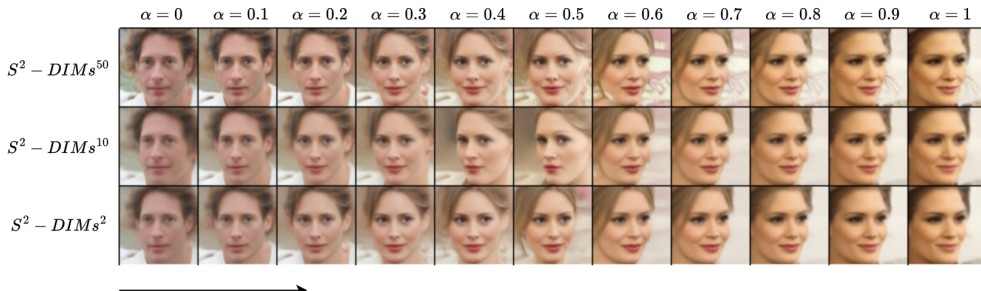

Figure 6: Interpolation of samples from the $S^2$-DMs in 10 steps. The $\alpha$ represents the weight used for interpolation, moving from left to right.

## 5 RELATED WORK

Denoising Diffusion Probabilistic Models (DDPMs; Ho et al. (2020)) and Noise Conditional Score Networks (NCSNs; Song & Ermon (2019)) are notable for their sample quality, comparable to GANs. While DDPMs optimize a variational lower bound, NCSNs target score matching over a Parzen density estimator. Both models employ a denoising autoencoder across noise levels and use Langevin dynamics for sampling. The shared approach requires multiple iterations for optimal sample quality. Recent advancements aim to decrease DDPMs' inference steps through dynamic SDE solvers and programming algorithms. However, challenges like the disparity between log-likelihood reduction and FID(Heusel et al. (2017), Szegedy et al. (2016)) remain in some models.

DDIMs(Song et al. (2020a)) emerges as an implicit generative model, wherein samples are uniquely defined by latent variables. This lends DDIMs properties akin to GANs and invertible flows, including the capability to produce semantically meaningful interpolations. Conceived from a purely variational standpoint, DDIMs sidesteps the constraints of Langevin dynamics, potentially explaining its superior sample quality compared to DDPMs in fewer iterations. The sampling paradigm of DDIMs also echoes the concepts found in neural networks with continuous depth. Additionally, other innovative methods have been introduced to further refine DDPMs sampling, such as reverse SDEs with unique coefficients, "corrector" steps, and probability flow ODEs. As the exploration of efficient sampling in diffusion models continues, our research stands on the shoulders of these pioneering works, aiming to push the boundaries of what's achievable in generative models.

## 6 CONCLUSION AND FUTURE WORK

We propose the Skip-Step Diffusion Models, a diffusion model that achieves higher quality samples solely by adding skip-step information during the training process, with no modifications required in the sampling procedure. We demonstrate how to incorporate skip-steps into the loss function during training and how to determine the weight between the newly added loss and the original loss function. Our results qualitatively and quantitatively show that the $S^2$-DMs significantly enhance the sample quality of image generation. Our approach successfully explores adding skip-step information to the training process, allowing the model to reach a symmetrical state and consequently achieve improved sample quality.

Our findings found a new direction for future research. The asymmetry between the training and sampling procedures of diffusion models can be studied, ensuring that the trained models align more closely with the sampling process. This results in obtaining high-quality samples, ensuring that even with fewer sampling steps, the model still generates high-quality outputs. Perhaps this presents an effective solution to the challenge diffusion models face in balancing sampling speed and sample quality. In the future, we will continue to research how to incorporate better skip-step information, ensuring the model aligns even more with its sampling process, and achieving higher quality samples in fewer sampling steps.

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

## A    EXPERIMENAL DETAILS

In this section, we include more details about the training and sampling of the $S^2$-DMs. All the experiments are run on two NVIDIA A100 GPUs.

### A.1    TRAINING

Our experiments were conducted on the CIFAR10 and CelebA. Since the original size of the CelebA dataset is not 64x64, we followed Song's approach by first center-cropping the CelebA images and then resizing them to 64x64 dimensions. During model training, we set the batch size to 128 and employed the Adam optimizer. On the CIFAR10 and CelebA datasets, we iterated for 600K and 400K times respectively, even though the typical number of iterations stated is 800K/600K. To investigate this effect, the main text also showcases experimental results from training for 800K/600K iterations. Finally, the images generated by the model were compared with a dataset of 50,000 images for FID calculation.

To ensure the repeatability of our experiments, we uniformly adopted the random seed 1234 from Song's repository by default. Additionally, on the CIFAR10 dataset, we refrained from utilizing multi-threading, guaranteeing that the reproducibility of the experiments would not be compromised by hardware randomness.

We quantitatively investigated the training overhead of the model. All results were measured on two NVIDIA A100 graphics processors. In Table 3, we report the time consumption and memory usage for each iteration on the CIFAR10 and CelebA datasets. As can be observed, the introduction of $L_{skip}$ increased the training overhead of the model. However, it did not significantly extend the overall training time. We believe that compared to the substantial performance improvement, the added overhead is acceptable.

Table 3: Training time (s) per iteration on the $S^2$-DMs.

| Dataset/Model | CIFAR/DDIMs | CelebA/DDIMs | CIFAR/$S^2$-DMs | CelebA/$S^2$-DMs |
|:---:|:---:|:---:|:---:|:---:|
| Time per iter(s) | 0.0025 | 0.0048 | 0.0031 | 0.0056 |
| Memory per GPU(G) | 4.83 | 15.55 | 4.83 | 15.56 |

### A.2    SAMPLING

In Table 4, we showcase the time required for the $S^2$-DMs to sample with $\{10, 20\}$ steps on different datasets. We believe that within this range, the trade-off between performance and time cost is optimal.

Table 4: Sample total times (s) on the $S^2$-DMs. Sampling 50K samples, each iteration is 10K/2K in CIFAR10/CelebA.

| Datasets-samplesteps | CIFAR-10 | CIFAR-20 | CelebA-10 | CelebA-20 |
|:---:|:---:|:---:|:---:|:---:|
| Total Times(s) | 675 | 1305 | 1483 | 2904 |

It's worth noting that when we trained and sampled the original DDIMs model on CelebA using mixed-precision training, we encountered issues related to gradient explosion. However, this problem did not arise when employing the same mixed-precision training with the $S^2$-DMs. We plan to investigate this issue further. For now, we believe it leans more towards engineering and hardware-related challenges.

In Table 5, we provide detailed numerical results of ablation experiments with different values of $skip$. The best results are highlighted in bold. From the table data, it's evident that with smaller sample steps, the larger the $skip$, the better the model performs, as it gets closer to the symmetric interval at this point.

Table 5: FID scores for the stpe ablation on CIFAR10 and CelebA. The impact of skip steps on the model was examined by varying the skip values among {50, 10, 2} based on DDIMs.

| Models \ # samplesteps S | 10 | 20 | 50 | 100 | 200 | 400 | 1000 |
|---|---|---|---|---|---|---|---|
| CIFAR10(32×32) | | | | | | | |
| $S^2$-DMs$^{50}$ | **8.01** | **6.44** | 6.86 | 7.31 | 7.65 | 7.92 | 8.19 |
| $S^2$-DMs$^{10}$ | 15.63 | 9.88 | **6.75** | **5.61** | **4.87** | **4.30** | **4.21** |
| $S^2$-DMs$^2$ | 17.92 | 11.00 | 7.34 | 5.85 | 5.06 | 4.37 | 4.26 |
| Celeba(64×64) | | | | | | | |
| $S^2$-DMs$^{50}$ | **6.41** | **3.99** | **3.99** | 4.73 | 5.57 | 6.34 | 6.62 |
| $S^2$-DMs$^{10}$ | 11.97 | 8.12 | 5.29 | **4.18** | **3.65** | **3.25** | **3.13** |
| $S^2$-DMs$^2$ | 12.43 | 8.73 | 6.00 | 4.80 | 4.13 | 3.77 | 3.71 |

### A.3 FEWER SAMPLESTEPS

Given the astonishing performance of the model with a skip value of 50 at fewer steps, we decided to try even fewer sampling steps. We selected some images for comparison with DDIMs (as shown in Figure 7 and 8). We are satisfied with the generated results, and they still outperform DDIMs.

## B EXTENDED SAMPLES

We provide extended samples of the $S^2$-DMs trained on CIFAR10 and CelebA. In order to demonstrate the effects of different sampling methods under different settings, we randomly selected several images for display among the images generated by the model.

### B.1 CIFAR10

We visualize the samples produced by different methods. In Figure 9 to Figure 12, we provide samples from models on CIFAR10 in 10 steps.

### B.2 CELEBA

In Figure 13 to Figure 16, we provide samples from models on CelebA in 10 steps.

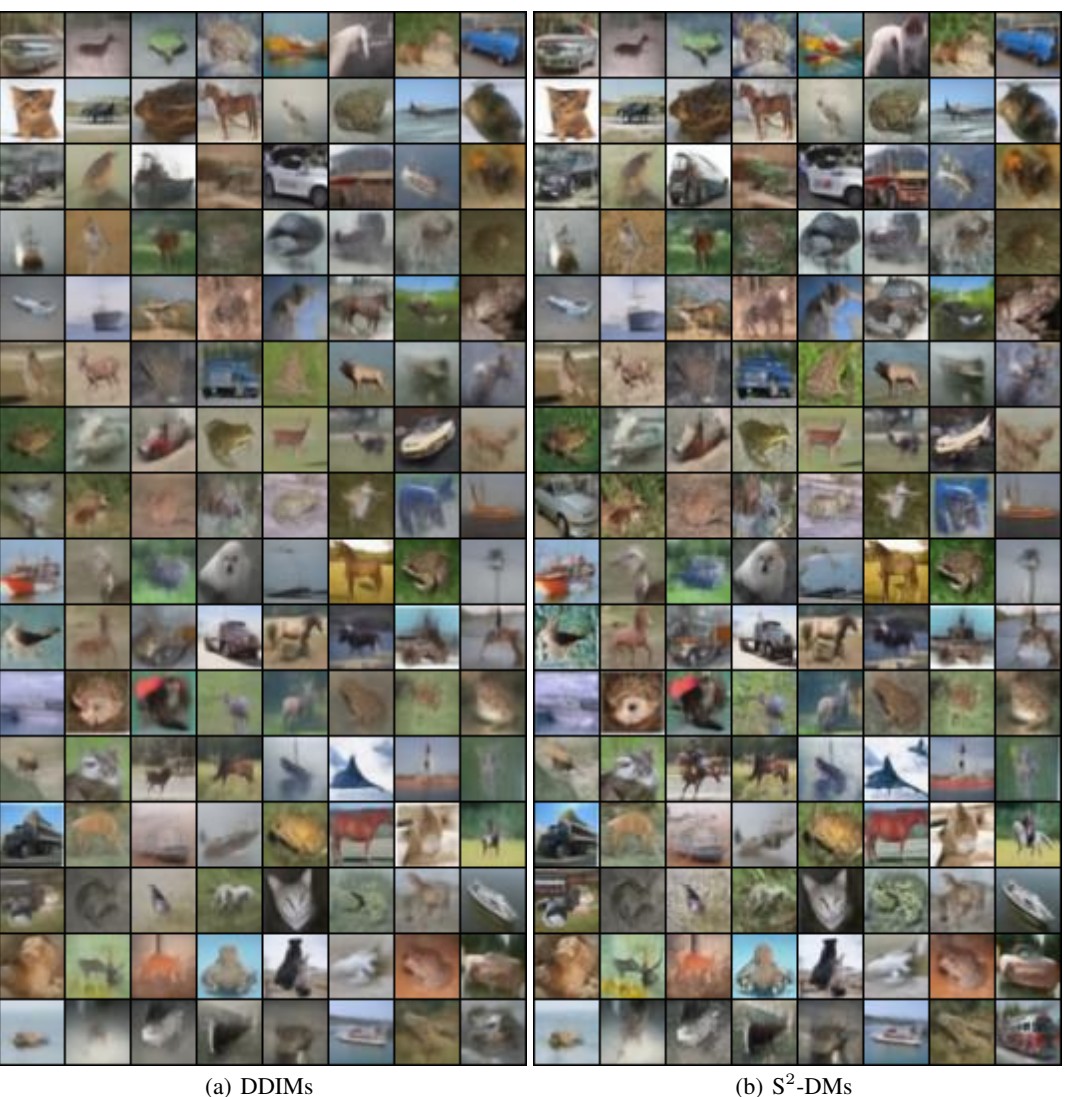

(a) DDIMs
(b) S$^2$-DMs

Figure 7: Sampling results for DDIMs and the S$^2$-DMs$^{50}$ on CIFAR10 in 5 steps.

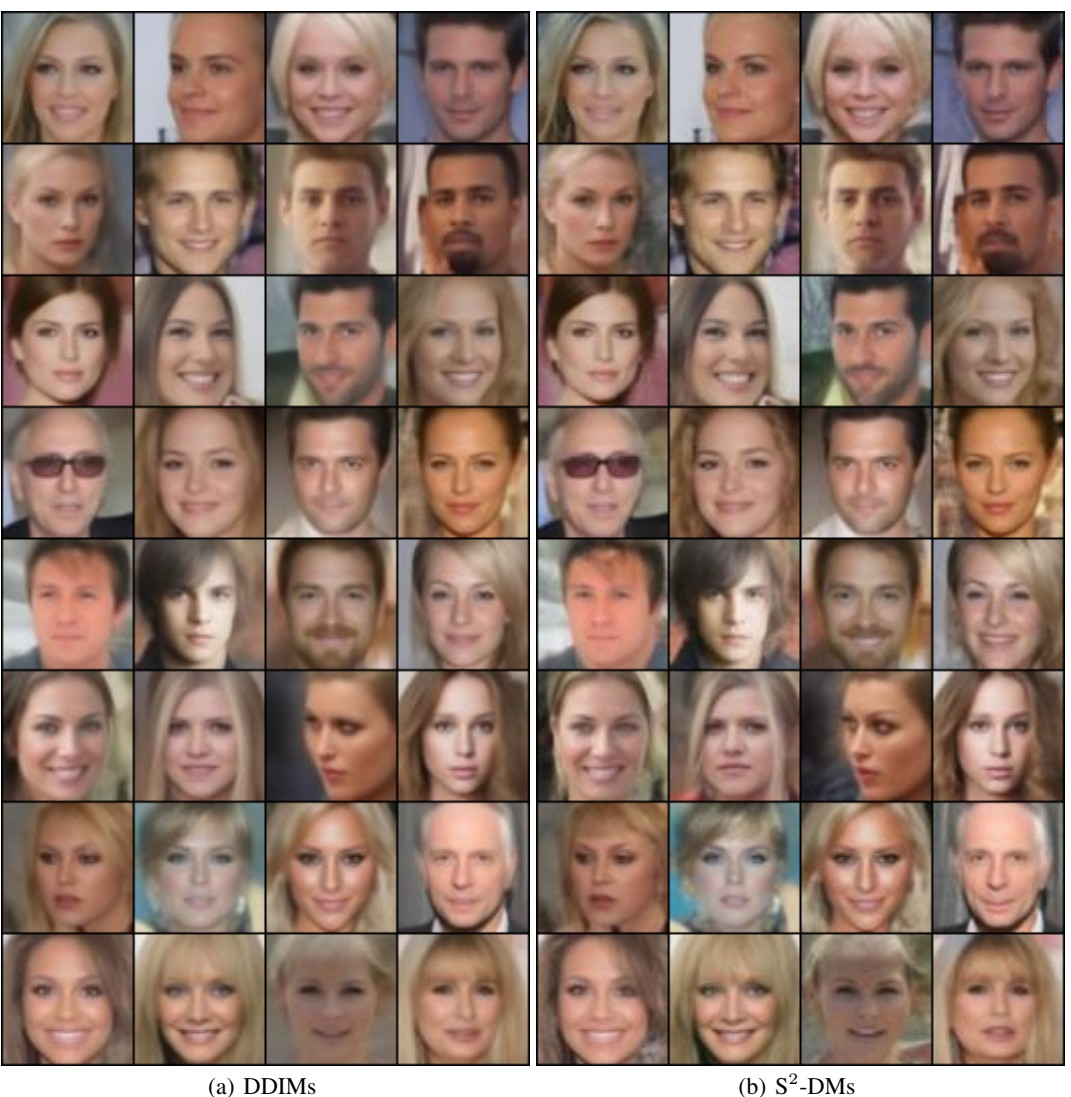

(a) DDIMs     (b) S$^2$-DMs

Figure 8: Sampling results for DDIMs and the S$^2$-DMs$^{50}$ on CelebA in 5 steps.

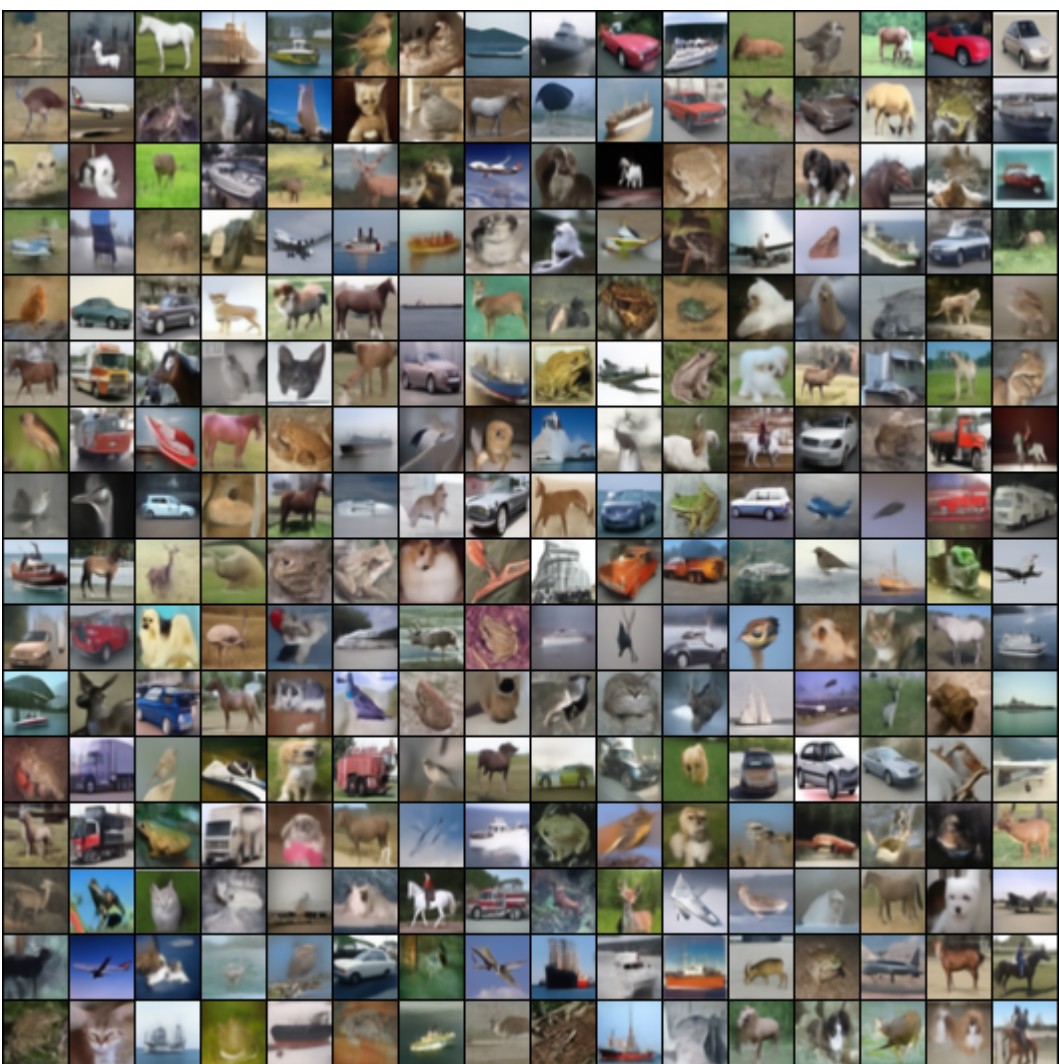

Figure 9: More samples from the CIFAR10 with the $S^2$-DMs$^2$ in 10 steps. FID=17.92.

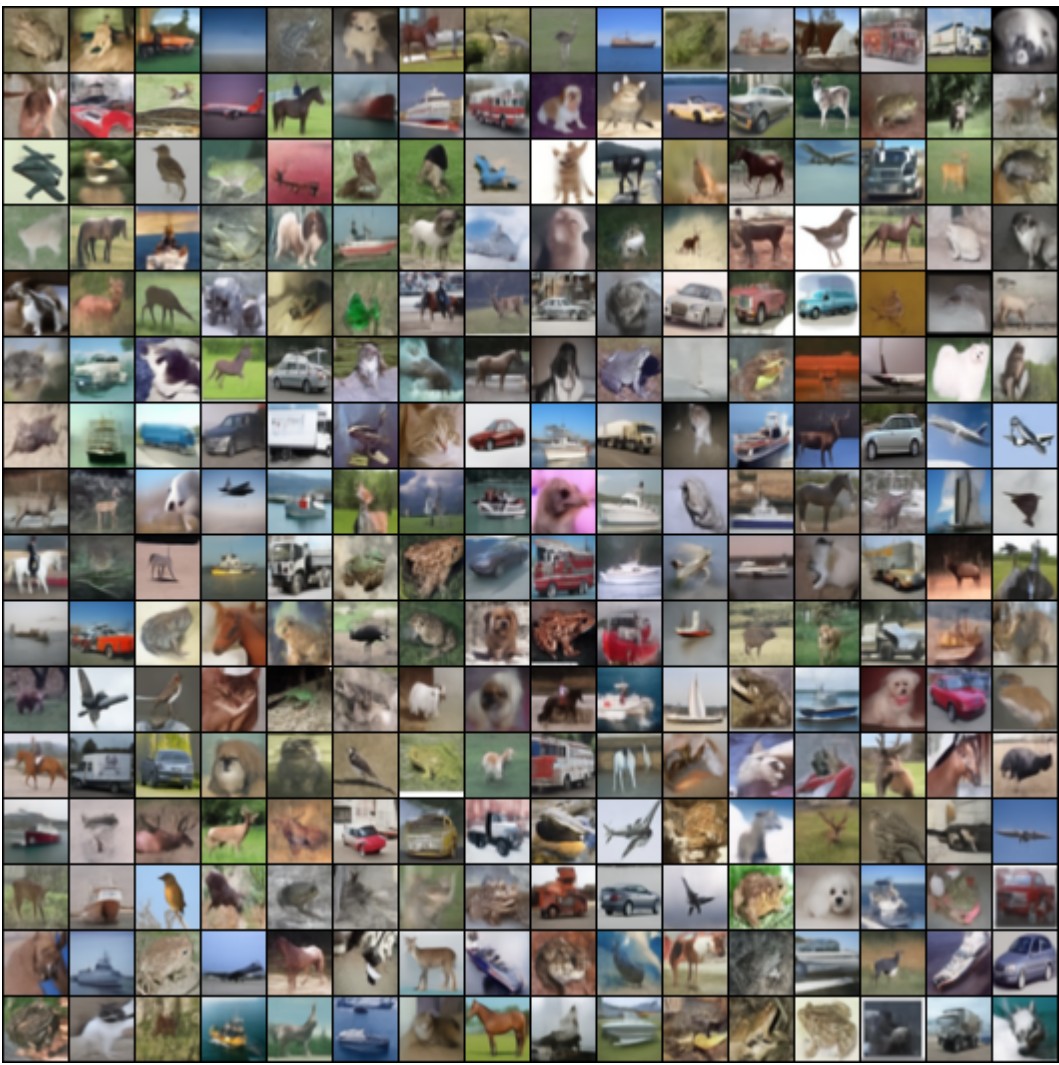

Figure 10: More samples from the CIFAR10 with the $S^2$-DMs[10] in 10 steps. FID=15.63.

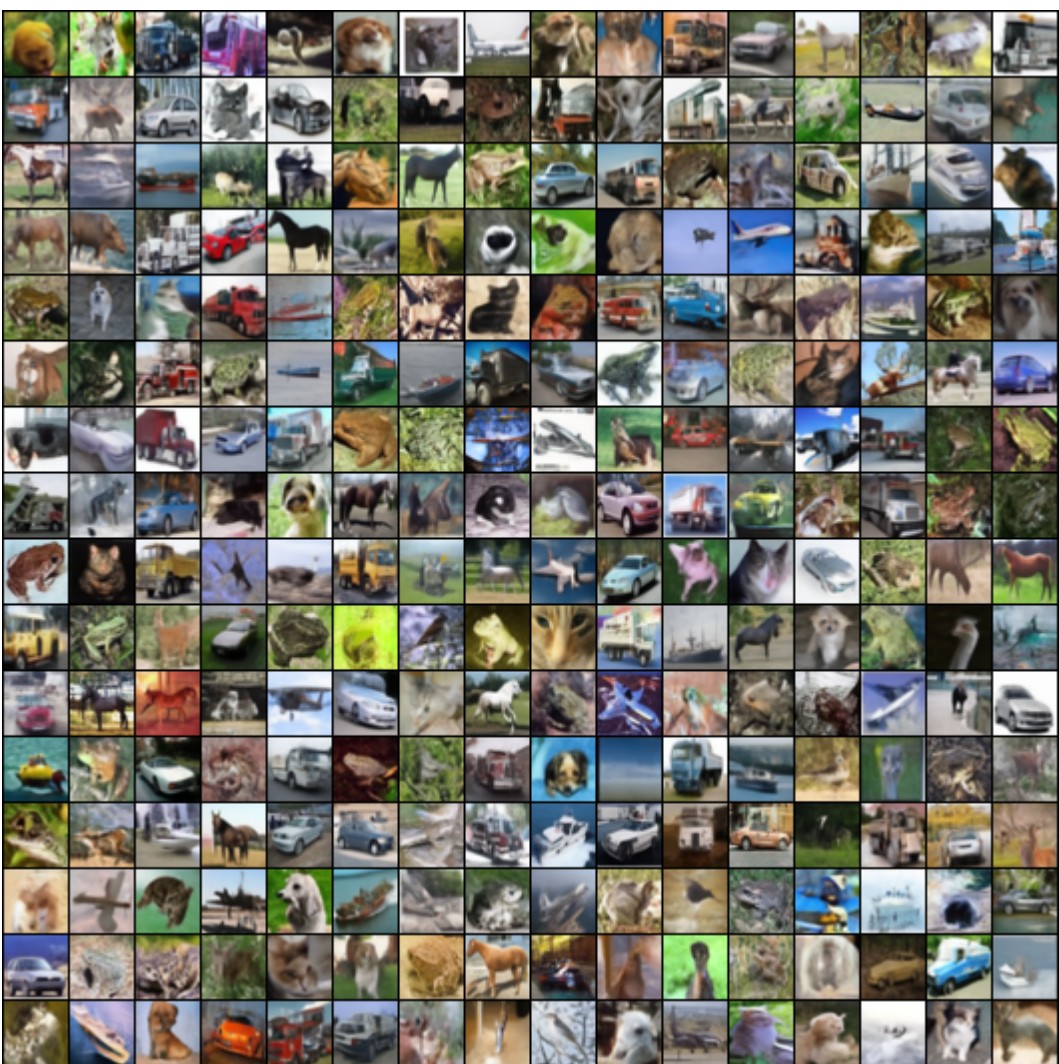

Figure 11: More samples from the CIFAR10 with the $S^2$-DMs$^{50}$ in 10 steps. FID=8.01

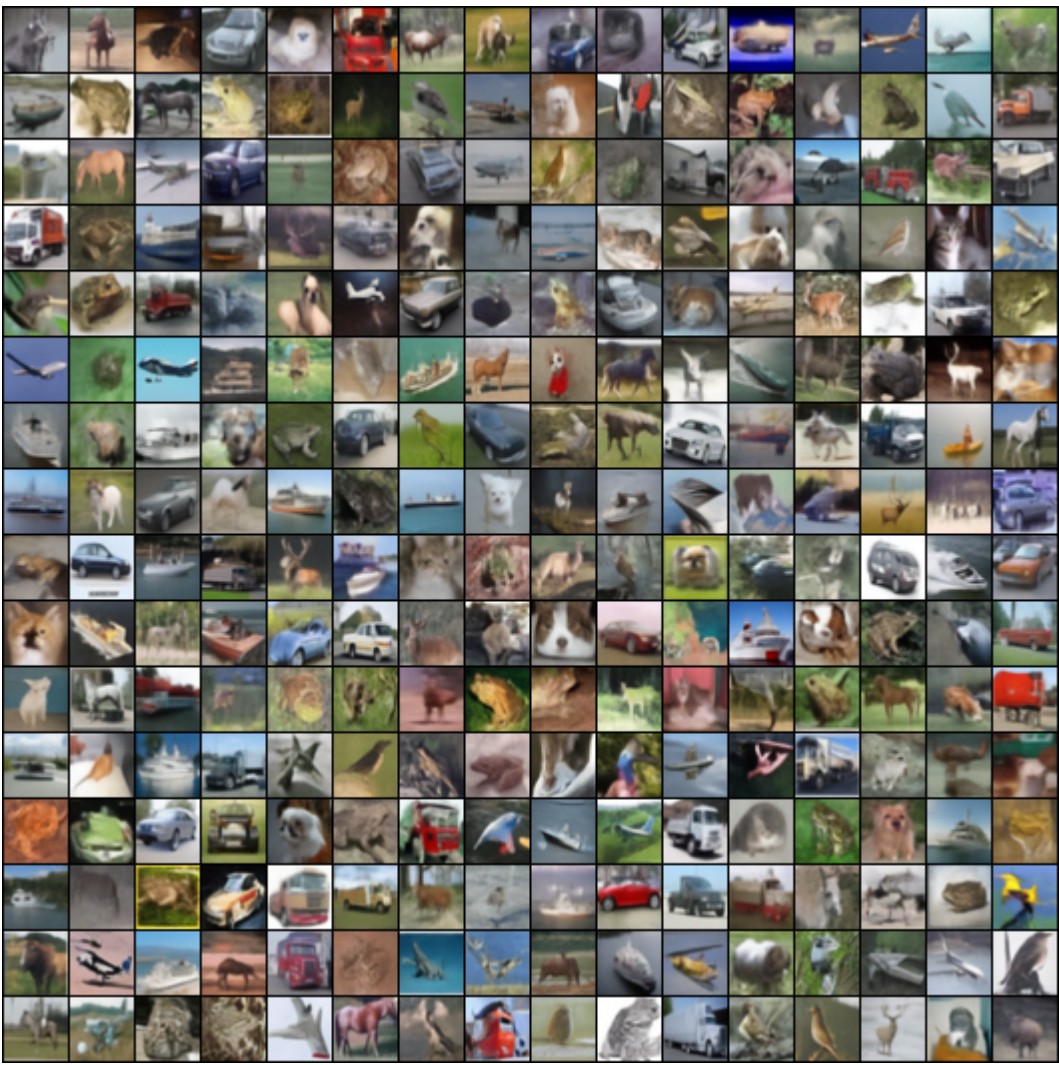

Figure 12: More samples from the CIFAR10 with the $S^2$-DMs$^{10}$(PNDMs) in 10 steps. FID=12.01.

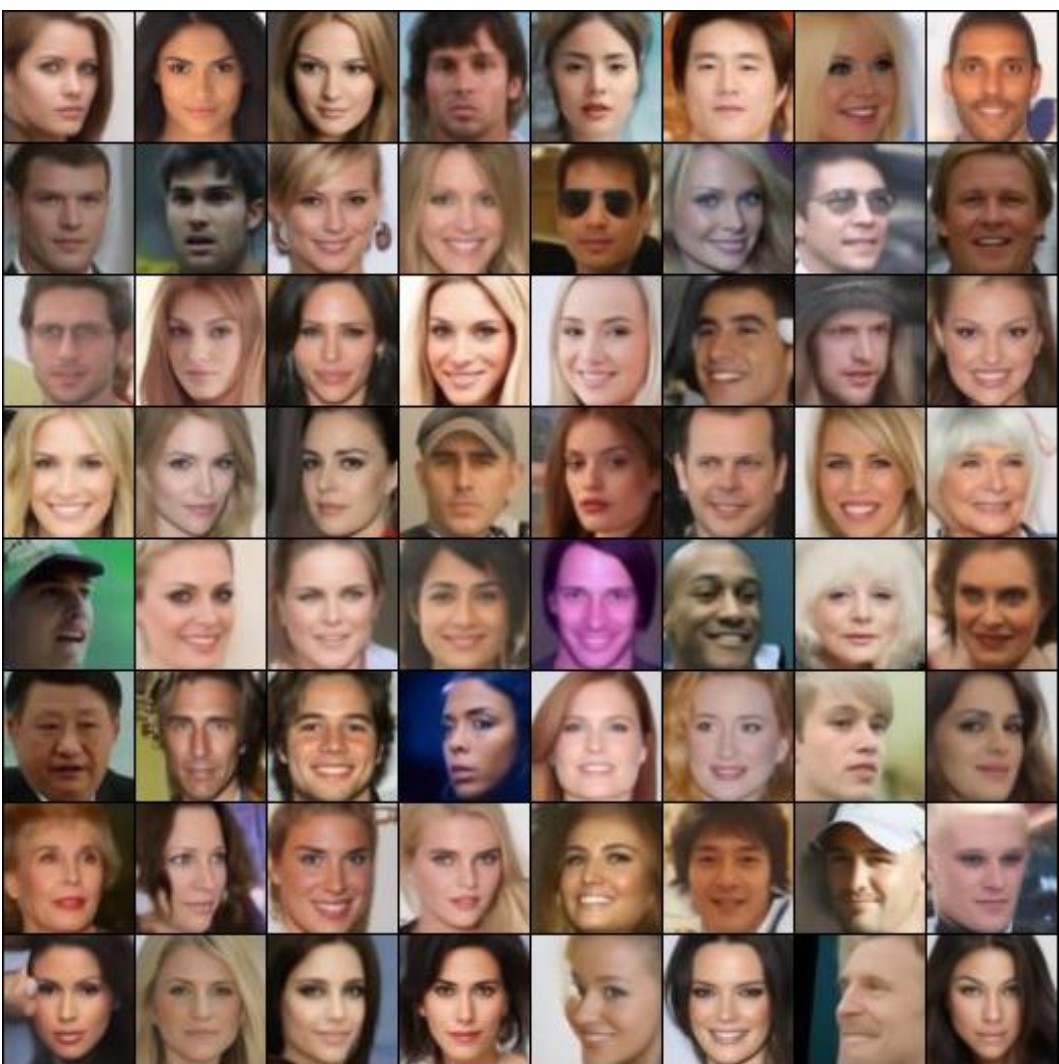

Figure 13: More samples from the CelebA with the $S^2$-DMs$^2$ in 10 steps. FID=12.43.

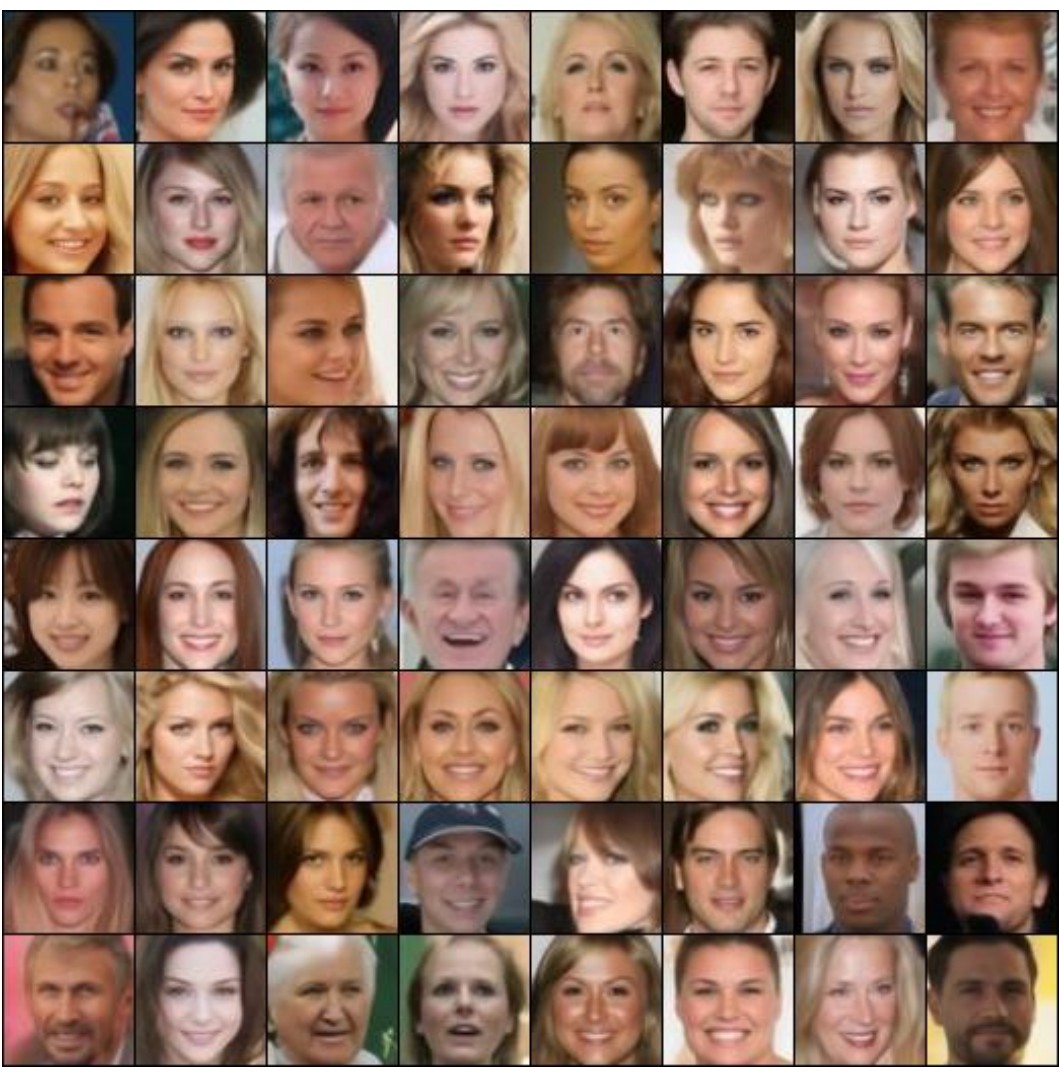

Figure 14: More samples from the CelebA with the $S^2$-DMs$^{10}$ in 10 steps. FID=11.97.

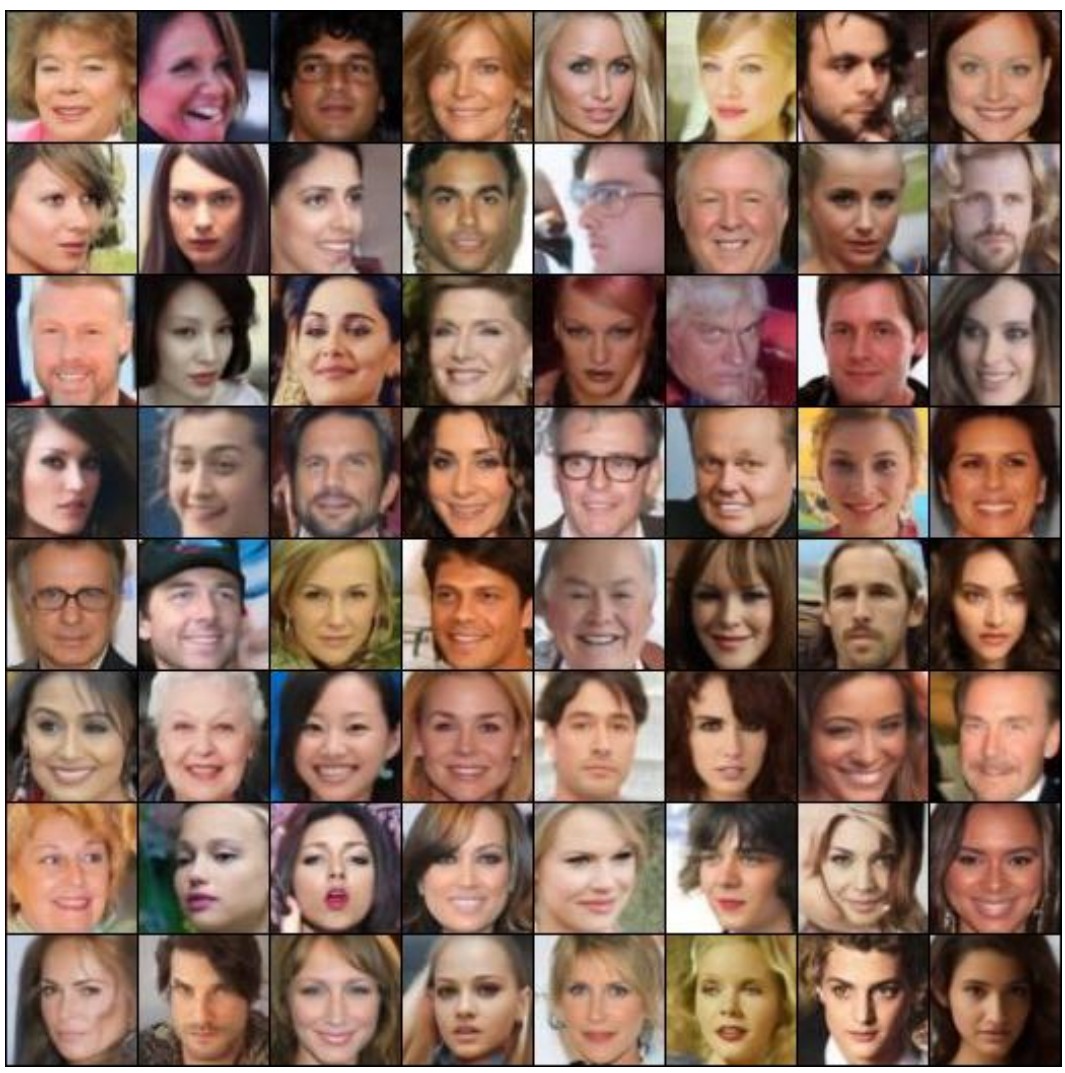

Figure 15: More samples from the CelebA with the $S^2$-DMs$^{50}$ in 10 steps. FID=6.41.

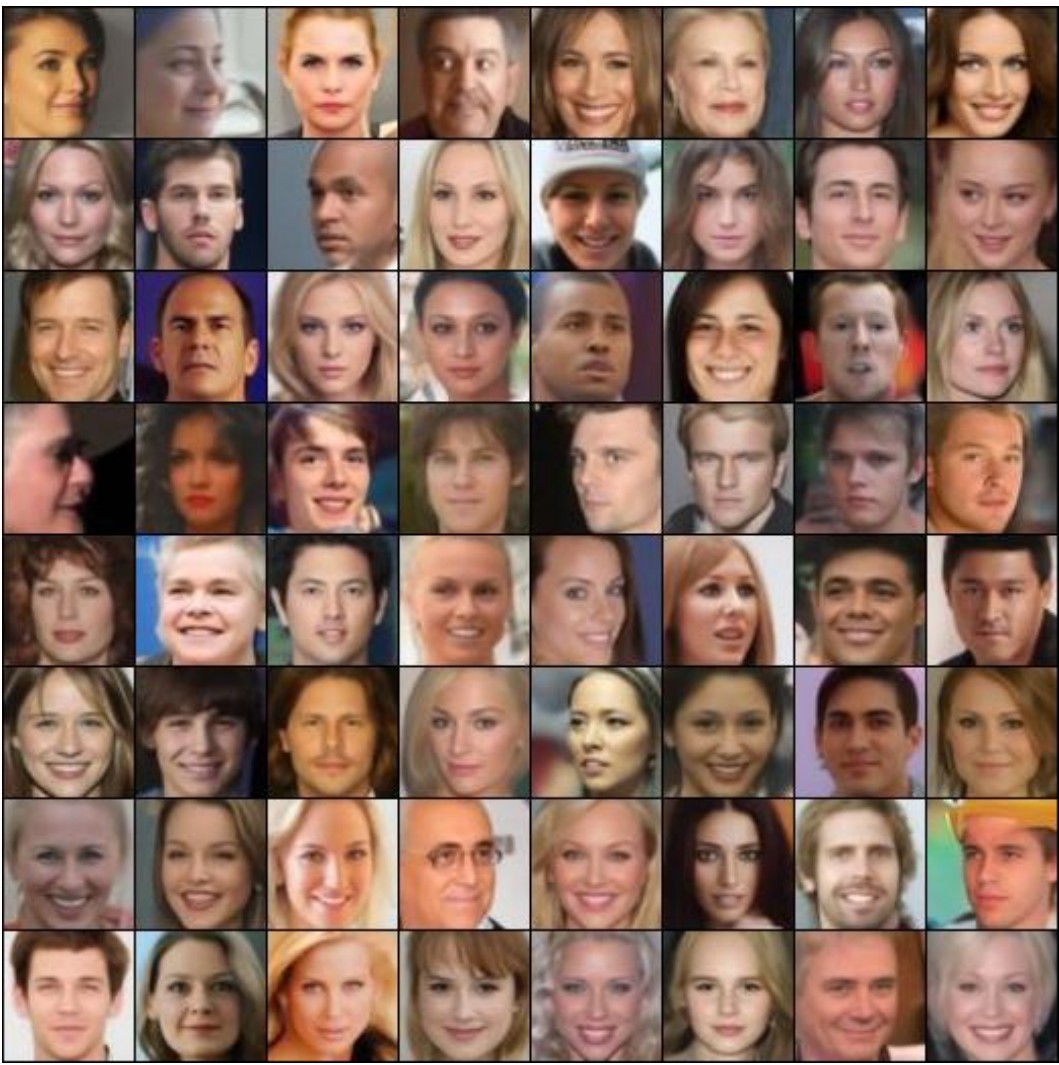

Figure 16: More samples from the CelebA with the $S^2$-DMs$^{10}$(PNDMs) in 10 steps. FID=11.40.

