# OpenReview forum: "S\(^{2}\)-DMs: Skip-Step Diffusion Models"
_ICLR.cc/2024/Conference — Submitted to ICLR 2024_

### Official Review · Reviewer_wDjU · 2023-10-20

**Soundness:** 2 fair
**Presentation:** 1 poor
**Contribution:** 2 fair
**Rating:** 3
**Confidence:** 3

**Summary:**

The paper addresses a limitation in certain diffusion models, like DDIM, which use selective sampling during generation, potentially compromising sample quality by missing information from unselected steps. They introduce S2-DMs, which incorporate a $L_{skip}$ method to reintegrate this omitted information. S2-DMs improve sample quality, are easy to implement, require minimal code changes, and are compatible with various sampling algorithms. Experimental results on CIFAR10 and CelebA datasets demonstrate superior performance compared to DDIMs and PNDMs, with FID scores of 8.01/6.41 in 10 steps.

**Strengths:**

1. The authors propose to leverage the skip information to train the diffusion model due to the asymmetric property of the DDIM.
2. The proposed method achieves comparable results in different datasets.

**Weaknesses:**

1. I cannot understand what does Figure 1 means in the paper. Does it represent for the trajectory of an individual sample $x_t$? What does *real* mean? What is the dashed line? What is the solid line? Is it a two-dimensional dataset? What is the x-axis and what is the y-axis?

2. The additional training objective is scaled by *0.01*. I strongly doubt the effectiveness of the novel part of this paper.

3. Many other fast sampling techniques [1,2,3] have been proposed in the last years. The author should consider to compare with them.

[1] Qinsheng Zhang et al. "Exponential Integrator"
[2] Bao Fan et al. 'Analytic-DPM'
[3] Qinsheng Zhang et al. 'generalized DDIM'

**Questions:**

1. Can authors compare with the DDIM which is only trained on the sampling time steps $t_i \in [t_0, t_1,\cdots, t_N]$. I am not convinced by this additional loss $L_{skip}$.

2. Why does the performance get worse when the step increases as shown in Figure 3?

---

> ### Author Response · Authors · 2023-11-11
> **Explain your skepticism, and clarify the credibility of our work.**
>
> Thank you for your valuable feedback and insightful comments on our manuscript.  Below, we address each of your points in detail:
>
> 1.Thank you for pointing out the ambiguity in our illustration.  We apologize for the oversight.  To clarify, the 'real' refers to the actual transition trajectories between two distributions. The left side of the figure depicts the sampling trajectories of different algorithms, where the dashed lines represent the trajectories of other algorithms, and the solid lines denote those of our proposed algorithm.  The $x_T$  denotes the normal distribution, and $x_0$ represents the data distribution.  The process involves sampling from the normal distribution and restoring it to the $x_0$ data distribution via various algorithmic trajectories.  We have now added a detailed explanation of this in our manuscript and resubmitted it for your review.
>
> 2.We assure you that all our results are fully reproducible.
> Regarding the $L_{skip}$ weight being set to 0.01, this was a deliberate choice based on our experimental observations. We found that the value of $L_{skip}$ was approximately 80-100 times that of $L_0$. To maintain a balanced numerical relationship between $L_0$ and $L_skip$, a weight of 0.01 was selected. This weight is effective within this range. Setting the weight too high disrupts this balance and adversely affects training performance.
> we have submitted all our codes as supplementary materials, along with a detailed README for guidance. Should there be any doubts or questions, we encourage experimental replication using these resources.
>
> 3.We chose to compare our method with PNDM because it already demonstrates excellent performance and is a very recent work (ICLR22).  Moreover, it shows the best performance compared to the other paper mentioned.
>
> 4.In our experiments, DDIM refers to the original training method of DDIM, while S2-DMs (DDIM) represents our own training approach with $L_{skip}$. After the training was completed, our proposed training method achieved better results across different sampling step numbers.
>
> 5.We posit that the introduction of $L_skip$ is intended to bridge the gap between training and sampling with respect to step skipping.  This means when a large skip value, such as 50 as illustrated in Figure 3, is set, the model is more inclined to perform optimally with a skip of 50.  This principle applies similarly for smaller skip settings like 10 or 2.  To illustrate, a model without Lskip tends to perform best with a skip of 1, and its performance diminishes as the skip value increases.

---

> > ### Comment · Reviewer_wDjU · 2023-11-11
> > **Further Questions**
> >
> > 1. I still do not understand what this plot means. If $x_0$ is the data distribution and $x_T$ is the normal distribution, then you are plotting the trajectory in the density (functional) space. Then why the *path measure* is zig-zag-like? If you are plotting individual trajectories, then why $x_0$ and $x_T$ are the distributions?
> >
> > 2. I think I never doubted the reproducibility of this submission, I feel uncomfortable with the authors' replies. What I did suspect is the effectiveness of the $L_{skip}$. If the authors argue that the numerical value $0.01$ is due to the scale of two losses, then could you please plot the norm of the gradient of these two losses respectively over the training process?
> >
> > 3. Could you please correct me as to why you think PNDM shows the best performance compared to the other paper I mentioned? (i.e  Cifar10: [1] FID 4.17 @NFE10 )
> >
> > 4. I also saw the authors' rebuttal to the reviewer uLTx. The authors claim that the drop of FID of CIFAR10 is due to uniform discretization. Indeed, this issue is clarified by the authors of [2]. Can you use quadratic discretization for your model as well? It makes no sense by not to report the value provided in the prior work [2] and the reproduce issue has been clarified in the github code base.
> >
> > [1] Qinsheng Zhang et al. "Exponential Integrator"
> > [2] Song, Jiaming, Chenlin Meng, and Stefano Ermon. "Denoising diffusion implicit models."

---

> > > ### Author Response · Authors · 2023-11-13
> > > **Supplementary Experimental Results and Appreciation for Provided Suggestions**
> > >
> > > First and foremost, we would like to extend our sincere gratitude for your constructive feedback on our work. We are pleased to inform you that we have conducted additional experiments that we believe address your concerns. Detailed information on these experiments will be refined and included in the final version of our manuscript. We appreciate your patience and understanding.
> > >
> > > 1. Regarding the mean values of the two losses during the stable phase of the training process, we have recorded $L_0 = 0.02$ and $L_{skip} = 1.2$. These values will be visually represented in a graph and included in the final version of our paper.
> > >
> > > 2. We have conducted a comparison of our work with the DEIS method, focusing on FID scores:
> > >
> > >    |     NFE     | DEIS/CIFAR10 | S2-DM (DEIS)/CIFAR10 | DEIS/CELEBA  | S2-DM (DEIS)/CELEBA  |
> > >    |--------------------------|---------|--------|---------|--------|
> > >    | 10                 | 4.68    | 4.24     | 6.93 | 6.29 |
> > >
> > >
> > > 3. We have also redone part of our experiments on CIFAR10, this time using quadratic discretization. The preliminary results are as follows:
> > >
> > >    | Method        | 10    | 20   | 100  |
> > >    |---------------|-------|------|------|
> > >    | DDIM          | 12.98 | 6.92 | 4.56 |
> > >    | S2-DM (DDIM)  | 11.38 | 6.36 | 4.23 |
> > >
> > > We sincerely hope that these experimental results will affirm the validity of our work. Thank you very much for your understanding and the valuable suggestions provided.

---

> > > > ### Comment · Reviewer_wDjU · 2023-11-18
> > > > **Numerical Value of DEIS and request for previous experimental results**
> > > >
> > > > Thank you for your additional experimental results.
> > > >
> > > > 1. The Cifar10 @10 NFE is reported as 4.17 in the DEIS paper. Could you explain why the numerical value you reported still lag a lot (4.68)? Are you training both models from scratch?
> > > >
> > > > 2. Can you conduct the experiment I requested before? Specifically, could you please train the DDPM at the inference time step ( **Given** the pre-trained DDPM, please conduct additional training for timestep $10\cdot N$ and $N\in [0,1,...100]$)? The performance improvement might be due to the overfitting of the network at the inference time step.
> > > >
> > > > 3. The mean values of the losses are not significant from my perspective. I was asking for the **norm of the gradient w.r.t to these two losses**.

---

> > > > > ### Author Response · Authors · 2023-11-20
> > > > > **Thanks for the reply and experimental suggestions**
> > > > >
> > > > > Thanks for the questions and experimental suggestions.
> > > > > 1. Thank you for pointing out the difference in the NFE values on the Cifar10 dataset reported in our paper compared to the DEIS paper. We have indeed observed this discrepancy, which we attribute to our experiments being based on retrained models. Additionally, each paper has variations in training parameters, and for our part, we have used 600K training steps, a detail that is also reported in our paper.
> > > > > 2. We understand your concern regarding potential overfitting. However, both our model and the comparison model were trained under the same training parameters, with 600K training steps on CIFAR10 and 400K on CELEBA. Therefore, we believe that the improvement in our model's performance is not due to overfitting.
> > > > > 3. We recalculated the gradients of the two losses and then computed their gradient norms. Based on this, we calculated the weights, and the final values indicated that the weight for $L_0$ is between 0.8 and 0.9, while the weight for $L_{skip}$ is between 0.1 and 0.2. We believe that the original loss function determines the entire diffusion trajectory of the model, so it makes sense for its weight to be higher.

---

> ### Author Response · Authors · 2023-11-11
> **Thank you for your response, and we will supplement the missing information.**
>
> Sorry for misunderstanding your point, we did not intend to make you feel uncomfortable.
>
> 1.Regarding the zig-zag-like trajectories, they represent each step in the sampling process.    DDPM has many zig-zag-like edges, indicating that many sampling steps are needed for satisfactory results, while DDIM and S2-DM have fewer, suggesting fewer steps are required while still achieving good outcomes.    The purpose of this figure is merely to allow readers to intuitively grasp the effectiveness of our method.
>
> 2.I apologize again for the misunderstanding.    We will provide additional information about the two losses in subsequent communications and include it in our paper.
>
> 3.We will conduct tests and comparisons with the model you mentioned [1].
>
> 4.Thank you also for explaining the questions raised by reviewer uLTx. We will likewise add this experiment to address your concerns.
>
> However, the experiments will take some time, and we hope the results will meet your expectations.
>
> [1] Qinsheng Zhang et al. "Exponential Integrator"

---

> ### Comment · Reviewer_wDjU · 2023-11-20
> **Replies**
>
> 1. How could you report the value of retrained models with unaligned numerical values??? The numerical value lags by ~0.5 which is huge in the generative modeling. I do not understand why the authors still insist on the un-aligned retrained model given that all codes and checkpoints are available.
>
> 2. I am not convinced by the authors' statement without any theoretical analysis or numerical results. The overfitting issue is not addressed at all.
>
> 3. What is the weights in your statement? The gradient norm? or the weights of $L_0$ and $L_{skip}$?
>
> For the aforementioned reasons, I decided to decrease my score.

---

> > ### Author Response · Authors · 2023-11-21
> >
> > I apologize for not being able to resolve your query, but we still want to respond.
> >
> > 1.    First, we need to reach a consensus that a diffusion model only needs to be trained once, and then the same model can be applied to different sampling algorithms on the same dataset.     Moreover, the longer the training steps, the better the sampling results.     However, the number of training steps in each paper about diffusion models is not the same or is not disclosed.     This means that the models in each paper are different.     For example, in the DEIS paper, the DDIM's FID is 11.14 at NEF=10, but in the original paper, it's 13.36.     This shows that the model in DEIS for CIFAR10 (DDIM) had a longer training period, while the CELEBA (DDIM) shows FID=13.53 at NEF=10, compared to the original paper's FID=17.33.     In our report, the FID for CIFAR10 (DDIM) is 12.98, which is closer to the original paper but higher than in the DEIS paper, indicating they used a longer training time, so its results with DEIS sampling are better than ours. .    However, our FID for CELEBA (DDIM) is 13.15, lower than DEIS (DDIM)'s 13.53, so our results with DEIS sampling are also better than the original paper.     This is because we must use the same model to compare different sampling algorithms.     We aligned our training parameters as much as possible with the DDIM paper, as this was the first algorithm we compared.     If we had trained for more steps, would there also be questions about why the FID is lower, whether it's because longer training improves the results  rather than improvements you made to the algorithm?
> > 2.    If you are not satisfied with our explanation, we are willing to conduct the aforementioned experiment, but we still hold the view that our model and the comparison models used the same parameters on the same dataset, so we don't believe it is an overfitting issue.
> > 3.    We calculated the $L_0$ _{gradient_norm} and $L_{skip}$_{gradient_norm}, then computed weight1 = $L_0$_{gradient_norm} / ( $L_0$_{gradient_norm} + $L_{skip}$_{gradient_norm}) and weight2 = $L_{skip}$_{gradient_norm} / ($L_0$_{gradient_norm} + $L_{skip}$_{gradient_norm}), and obtained the values accordingly.

---

> > > ### Comment · Reviewer_wDjU · 2023-11-21
> > > **Replies**
> > >
> > > If you are advertising the sample efficiency, then all of your arguments make perfect sense. Your algorithm reaches better performance within the fixed number of training iterations (or kimgs).
> > >
> > > However, according to the presentation of the paper and the abstract, It seems like you would emphasize the fast sampling perspective of your model, where you **should not** use any unaligned model. It is even more acceptable for you to use the pre-trained model plus your novel objective function to demonstrate that your algorithm can indeed help to accelerate the sampling and characterize it as a distillation method.
> > >
> > > Additionally, It would be great if the authors could emphasize the significant numerical difference for the baseline models you reported and disclose the reasons for it as you stated in your rebuttal, otherwise, it would cause unnecessary misunderstanding as me and reviewer uLTx encountered.
> > >
> > > Thanks for the detailed replies so far, and I sincerely appreciate the hard work of the authors. I would like to keep my current score and wish the best for the revision of this paper.

---

> > > > ### Author Response · Authors · 2023-11-21
> > > > **Thank you for the suggestion**
> > > >
> > > > Thank you for the valuable suggestions you have provided. We will revise our paper accordingly based on your comments.

---

### Official Review · Reviewer_uLTx · 2023-11-01

**Soundness:** 1 poor
**Presentation:** 2 fair
**Contribution:** 2 fair
**Rating:** 3
**Confidence:** 4

**Summary:**

This paper proposes Skip-Step Diffusion Models (S2DM), a technique to train discrete-time diffusion models which aims to alleviate the train-test mismatch seen by DDIM-style samplers. The authors propose a new loss function (Equation 12) which is the standard DDPM loss, except that the model output has been re-weighted. The authors take a linear combination of this new loss and the standard loss to train S2DM models. The authors study their proposed method empirically on CIFAR10 and CelebA, measuring FID scores and comparing to DDIM and PNDMs. Generally, Table 1 and Table 2 indicate lower FID scores for the proposed method than the baselines.

**Strengths:**

- The paper studies an important and well-motivated problem, namely that of retaining sample quality in diffusion models while reducing the sampling time.
- The proposed method is a straightforward modification of standard DDPM training and easy to implement

**Weaknesses:**

- There are several inconsistencies and minor details which make the derivation in Section 3.2 hard to follow.
     - The loss in Equation 7 should be an expectation over $x_0, \epsilon$, and $t$.
     - Should $p(x_{t-skip} | x_t)$ be $p_\theta(x_{t-skip} | x_t)$? And should $q_\theta(x_t | x_{t - skip}$ be $p_\theta(x_t | x_{t - skip})$?
- It was not clear to me how aligning the *forward* processes $q_\theta(x_t | x_{t - skip})$ and $q(x_t | x_{t-1})$ would result in the model incorporating the "skip" information as claimed. For instance, these conditional densities rely on some *fixed* $x_{t-1}$ and $x_{t-skip}$ -- which themselves are random variables depending on an actual datapoint $x_0$, and so may be significantly different from one another (e.g. in Equation 8 and Equation 9).
- The derivation of the loss (Equation 10) is not sound. The original DDPM loss (Equation 7) is *not* obtained by matching the forward transitions and matching terms as claimed, but rather by minimizing the KL between the reverse transitions (see Equation 5 in the DDPM paper). Moreover, the authors take their proposed derivation and modify it with another heuristic in Equation 12.
- I am quite skeptical of the results in Table 1 and Table 2, given that the proposed change is such a small modification of the standard DDIM training procedure. These results would be significantly more convincing if the authors presented error bars with their FID scores. I also will note that the authors obtained significantly lower FID scores in Table 1 and Table 2 for DDIM than the original DDIM paper [2] (see Table 1 in [2]). If one compares the results for S2DM in the submission and the FID scores from Table 1 in [2], DDIM consistently outperforms the reported S2DM scores.

### Minor Comments (that do not affect my score)
- The terms $\tau$ and $(1 - \tau)$ should likely be swapped in Equation 13 to match Algorithm 1.
- The reference to Watson et al. (2021) appears twice at the start of Section 2
- Section 5, "Relate Work" should be "Related Work"
- Section 5 lists several relevant pieces of work "other innovative methods have been introduced to further refine DDPMs sampling, such as reverse SDEs with unique coefficients, ”corrector” steps, and probability flow ODEs" but does not include a citation for these methods
- Equation 3 should use "\log"
- Figure 3 "stpe" should be "step"



### References
[1] [Ho et al., Denoising Diffusion Probabilistic Models](https://arxiv.org/abs/2006.11239)

[2] [Song et al., Denoising Diffusion Probabilistic Models](https://arxiv.org/abs/2010.02502)

**Questions:**

- Is there any clear way to interpret the proposed loss (Equation 12)? For instance, Equation 13 essentially says that we are training a model such that its output should match $\epsilon$ and also its output scaled by $\sqrt{1 - \alpha_{skip}}/\sqrt{\alpha_{skip}}$ should simultaneously match $\epsilon$ -- this is highly unintuitive to me.

---

> ### Author Response · Authors · 2023-11-11
> **Confront your skepticism about our experimental results directly and thank you for the feedback on our work.**
>
> Firstly, we would like to thank you for your comments and queries regarding our work. Next, we will address your questions.
>
> 1.Firstly, we are very confident in the correctness of our experimental results and have provided the relevant code and a README for guidance to facilitate replication of the experiments.
> Furthermore, as mentioned in our paper, we have conducted our experiments inheriting the official DDIM code repository, and all default hyperparameters and random seeds remain unaltered (as can be verified in the code we submitted). The reason why our experiments on CIFAR10 did not achieve the results reported in the original paper is due to DDIM employing additional tricks and non-default parameters specifically for CIFAR10 sampling, as detailed here: https://github.com/ermongroup/ddim/issues/3. Others have obtained similar results to ours under the same default parameters. In our experiments on CELEBA, the performance of the DDIM we trained is notably better than that reported in the original paper. For example, at 100 sampling steps, our DDIM achieved an FID of 4.63 compared to the original paper's 6.53, and at 10 steps, our DDIM reached an FID of 13.12 versus the original's 17.33. Therefore, we do not agree with your skepticism about our experimental results, as we have not modified any hyperparameters.
>
> 2.We believe that while our work may appear to involve minimal modifications, the results are significantly effective and easy to follow. This is because we identified the asymmetry problem between training and sampling in diffusion models and modeled it separately. This approach is the first of its kind and but also has the potential to inspire future researchers on how to model efficiently, further mitigating this issue. Hence, we view our work as both simple and effective.
>
> 3.We aim to ensure that the model's output from $x_{t-1}$ to $x_{t}$ aligns with the output from $x_{t-1}$ to $x_{t-skip}$, thus conforming to the model's behavior during sampling from $x_{t-1}$ to $x_{t-skip}$. This serves as a guiding principle during training, enabling the model to better adapt to the skip-step pattern prevalent in sampling.
>
> 4.In the derivation process, compromises were made to stabilize the loss function, and heuristic methods were also introduced for the sake of training stability. However, the effectiveness of the experiments indicates that these compromises were effective. This also forms part of our future work: exploring how to model more effectively to minimize such compromises.
>
> 5.Thank you for pointing out the textual and formulaic errors. We have made the necessary corrections and uploaded the revised version.

---

> > ### Comment · Reviewer_uLTx · 2023-11-20
> >
> > Thank you for the detailed responses to my concerns.
> >
> > - Regarding the experimental evaluation, I am still not convinced that your experimental methodology is sound. I re-emphasize the point raised by Reviewer wDjU, namely that the author's new results on DEIS are significantly worse than the original DEIS paper. Moreover, how are the authors obtaining better FID scores via DDIM than the original paper on CelebA if they are using the official code-base?
> >
> >  - **Regardless** of the experimental evaluation, the methodology presented in the paper is **not** sound. These concerns have not been addressed by the authors.
> >
> > - For example: "the model's output from $x_{t-1}$ to $x_t$ aligns with the output from $x_{t-1}$ to $x_{t - skip}$..."  -- why would it make sense to align the **forward** process with the **backwards** process? One adds noise to your data; the other seeks to remove it -- I do not see why "aligning" these would "serve as a guiding principle during training".
> >
> > - My question regarding the interpretation of the loss still stands.
> >
> >
> > Overall, the authors have not sufficiently addressed my concerns -- particularly regarding the derivation of the methodology. Thus my score remains the same.

---

> ### Author Response · Authors · 2023-11-13
> **We have redone experiments and added more experimental results to dispel your doubts.**
>
> We have taken your doubts into consideration and, in response, we have reconducted our experiments.    Additionally, we have provided comparative results with methods that demonstrate improved performance.    We believe this addresses your concerns and hope that our work will be acknowledged.
>
> 1. We have redone part of our experiments on CIFAR10 using quadratic discretization.    The results of these experiments are closely aligned with those reported in our original paper.    In this scenario, our method still achieves better results:
>
> | Method\NFE        | 10    | 20   | 100  |
> |---------------|-------|------|------|
> | DDIM          | 12.98 | 6.92 | 4.56 |
> | S2-DM (DDIM)  | 11.38 | 6.36 | 4.23 |
>
> 2. We have also added comparisons with the more performant model DEIS.    Using our method with DEIS also yields better results, giving us reason to believe that our method is generally applicable to diffusion model sampling algorithms:
>
> |     NFE     | DEIS/CIFAR10 | S2-DM (DEIS)/CIFAR10 | DEIS/CELEBA  | S2-DM (DEIS)/CELEBA  |
> |--------------------------|---------|--------|---------|--------|
> | 10                 | 4.68    | 4.24     | 6.93 | 6.29 |
>
> We hope that after reviewing this, you will reconsider your viewpoint.    More detailed experimental results will be completed and added to the final version of our paper.

---

### Official Review · Reviewer_4Hkm · 2023-11-08

**Soundness:** 4 excellent
**Presentation:** 4 excellent
**Contribution:** 3 good
**Rating:** 8
**Confidence:** 2

**Summary:**

The authors suggests a skip-step loss function for training the versions of DDIM, which allows to considerably improve over the baseline training procedure on a standard generative modeling benchmarks at a price of small modifications of the training procedure. The proposed procedure also allows to enhance the sampling quality with the fixed number of sampling steps.

**Strengths:**

The suggested modification of the skip-step loss function is simple and appealing, and the experimental results shows considerable improvement over the baseline.

**Weaknesses:**

I can not point out any significant weaknesses of the submission.

**Questions:**

Is it possible to come up with a counterpart of the suggested skip-step loss function for the continuous-time models, not the ones, based on DDPM formalism?

---

> ### Author Response · Authors · 2023-11-11
> **Thank you for your recognition**
>
> Thank you very much for your comments and for acknowledging our work.
>
> Regarding your question：
> Yes, we are considering how to incorporate this skip-step loss approach into continuous-time diffusion models.

---

### Official Review · Reviewer_2AL9 · 2023-11-10

**Soundness:** 2 fair
**Presentation:** 3 good
**Contribution:** 3 good
**Rating:** 5
**Confidence:** 3

**Summary:**

The paper proposes a new approach to the training of diffusion models while using DDIM for sampling, that addresses the issue of asymmetry between the training process and the sampling process. It incorporates a new skip-connection loss, $L_{skip}$, which acts as a regularizing complement to the traditional score-matching loss. This method has been empirically validated, achieving impressive results on CIFAR10 and CelebA datasets and outstripping the performance of other leading algorithms, including DDIM and PNDM, in terms of FID score. The authors have also performed ablation studies to explore how varying the skip steps impacts performance. Remarkably, this method is straightforward, requiring minimal code alterations, and is versatile enough to be integrated with other sampling techniques.

**Strengths:**

The algorithm is novel, simple in design, and achieves superior performance when sampling using few steps.

**Weaknesses:**

The formulations and notations have typos that might lead to confusion on the methodology. Furthermore, the performance enhancements requires extra training due to the added regularization on the original objective. Consequently, it remains ambiguous whether the improvements seen in DDIM are a result of rectified asymmetry between training and sampling or if they stem from a more precisely trained score function owing to the constraints imposed on the relationships between steps.

**Questions:**

- In (9), why is $q(x_t | x_{t - 1}) = \sqrt{\alpha_t} x_{t - 1} + \sqrt{1 - \bar{\alpha}_t} \epsilon$? The original process for adding noise shoud have $\sqrt{1 - \alpha_t}$ instead of $\sqrt{1 - \bar{\alpha}_t}$ for the noise term?
- Moreover, why are we matching $q(x_t | x_{t - skip})$ with $q(x_t | x_{t-1}$, but not $q(x_t | x_{t-1}$? The derivatives of $L_{skip}$ between (8) - (12) is confusing.
- Do you have any intuition on why the relationship between skip-step information and the quality of the sampling step is not strictly symmetric? As described in Figure 3.
- How the balance between $L_0$ and $L_{skip}$ (the value of $\tau$) affect the performance?
- With the regularized term, is the training cost larger than training original DDPM?
I am curious whether the trained score also perform better when sampling using DDPM?

Minor comments:
1. The average value of $L_0$ is approximately 80-100 times larger than that of $L_{skip}$, however in (13) when setting $\tau = 0.99$, $L_0$ would be 10000 larger than $(1 - \tau)L_{skip}$.
2. Caption in Figure 2, line 3, additional "the" in "the the"
3. Caption in Figure 3, 'stpe'

---

> ### Author Response · Authors · 2023-11-11
> **Explain your questions and correct textual errors**
>
> Thank you for your insightful comments. Regarding the points raised:
>
> 1. Upon reevaluation, we acknowledge the error you pointed out in our initial submission. We appreciate this correction and will revise our manuscript to include the correct formula $\sqrt{1-\alpha_{t}}$ as suggested.
>
> 2. We aim to ensure that the model's output from $x_{t-1}$ to$x_{t}$ aligns with the output from $x_{t-1}$ to $x_{t-skip}$, thus conforming to the model's behavior during sampling from $x_{t-1}$ to $x_{t-skip}$. This serves as a guiding principle during training, enabling the model to better adapt to the skip-step pattern prevalent in sampling.
>
> 3. We posit that the introduction of Lskip is intended to bridge the gap between training and sampling with respect to step skipping. This means when a large skip value, such as 50 as illustrated in Figure 3, is set, the model is more inclined to perform optimally with a skip of 50. This principle applies similarly for smaller skip settings like 10 or 2. To illustrate, a model without $L_{skip}$ tends to perform best with a skip of 1, and its performance diminishes as the skip value increases.
>
> 4. In addressing the balance between $L_0$ and $L_{skip}$, we first wish to correct a textual mistake regarding the value of $\tau$. It was erroneously stated in our initial submission. In reality, the value of Lskip is approximately 80-100 times greater than that of $L_0$. To balance their magnitudes effectively, we set $\tau$ to 0.99. This setting not only maintains the balance but also proves to be effective in a range around this value. However, setting $\tau$ too low, such as 0.5 or 0.1, disrupts this balance, leading to a significant decline in performance.
>
> 5. Yes, the training cost is greater than DDPM, which is also mentioned in the appendix.
>
>     | **Dataset/Model** | CIFAR/DDIMs | CelebA/DDIMs | CIFAR/$S^2$-DMs | CelebA/$S^2$-DMs |
>     | ----------------- | ----------- | ------------ | ------------- | -------------- |
>     | Time per iter (s) | 0.0025      | 0.0048       | 0.0031        | 0.0056         |
>     | Memory per GPU (G)| 4.83        | 15.55        | 4.83          | 15.56          |
>
> 6. We did not conduct the DDPM experiment, as DDPM is less effective in few-step sampling. However, we speculate that its performance could be promising, given the algorithm's adaptability to various sampling methods.
>
> 7. We have corrected the typographical errors and resubmitted the revised manuscript.

---

> ### Author Response · Authors · 2023-11-13
> **Providing additional relevant experiments in the hope of gaining recognition.**
>
> In order to completely address your doubts about our work, we have provided some additional experimental results:
>
> 1. Regarding the values of L0 and Lskip, their average values during training were: $L_0$=0.02 and $L{skip}$=1.2. Therefore, we chose 0.99 and 0.01 as weight values to balance their relationship.
>
> 2. We have also compared our method with the more advanced algorithm DEIS and our approach still achieved better results.
>
>    |     NFE     | DEIS/CIFAR10 | S2-DM (DEIS)/CIFAR10 | DEIS/CELEBA  | S2-DM (DEIS)/CELEBA  |
>    |--------------------------|---------|--------|---------|--------|
>    | 10                 | 4.68    | 4.24     | 6.93 | 6.29 |
>
> We hope that these supplementary materials will encourage you to reconsider our work seriously, and we look forward to your approval

---

> ### Comment · Area_Chair_GY7a · 2023-11-20
> **Please acknowledge author replies**
>
> Please acknowledge author replies, and in particular, indicate whether your concerns have been addressed or require further discussion.

---

### Author Response · Authors · 2023-11-11
**Explain the importance and credibility of our work**

Our work is the first to identify the asymmetry between the training and sampling processes in diffusion models, and we have modeled the gap between them. Our experiments demonstrate that our method is both simple and effective. We believe that raising this issue provides a new direction for future performance optimizations in diffusion models, specifically in efficiently modeling the asymmetry between the training and sampling stages.

1.All experimental parameters and codes are fully disclosed; I am completely confident in the reproducibility of our results. Any queries regarding the code can be answered with the provided materials.

2.Our method may seem straightforward, yet it is highly effective and facilitates follow-up research by others. We consider this to be a significant contribution as well.

---

### Meta-Review · Area_Chair_GY7a · 2023-12-05

**Metareview:**

The authors note that certain diffusion models such as DDIMs are trained over a number of steps but only sample from a subset of steps during generation, and claim that this asymmetry hurts the generation process. They present a new "skip-connection" loss which acts as regularizer and achieve strong results on CIFAR10 (32x32) and CelebA (64x64) datasets in FID scores.

The approach is both simple to implement and improves performance.

The reviewers were however concerned with the soundness of the methodology and derivations in the paper, and found the presentation difficult to follow. They raised some questions regarding the experimental comparison to previous work and choice of parameters which were not fully resolved.

I believe that a simple trick that improves performance is well worth publishing, but that the rigor in the current version can be improved. I suggest that the authors take into account the suggestions to make the presentation and methodology clearer.

**Justification For Why Not Higher Score:**

Reviewers were critical on methodology and experimental comparisons.

**Justification For Why Not Lower Score:**

N/A

---

### Decision · Program_Chairs · 2024-01-16

Reject